# Structural basis of transcription inhibition by the DNA mimic protein Ocr of bacteriophage T7

**Fuzhou Ye[1], Ioly Kotta-Loizou[2], Milija Jovanovic[2], Xiaojiao Liu[1,3], David TF Dryden[4], Martin Buck[2], Xiaodong Zhang[1]***

[1]Section of Structural Biology, Department of Infectious Diseases, Faculty of Medicine, Imperial College London, London, United Kingdom; [2]Department of Life Sciences, Faculty of Natural Sciences, Imperial College London, London, United Kingdom; [3]College of Food Science and Engineering, Northwest A&F University, Yangling, China; [4]Department Biosciences, Durham University, Durham, United Kingdom

**Abstract** Bacteriophage T7 infects *Escherichia coli* and evades the host restriction/modification system. The Ocr protein of T7 was shown to exist as a dimer mimicking DNA and to bind to host restriction enzymes, thus preventing the degradation of the viral genome by the host. Here we report that Ocr can also inhibit host transcription by directly binding to bacterial RNA polymerase (RNAP) and competing with the recruitment of RNAP by sigma factors. Using cryo electron microscopy, we determined the structures of Ocr bound to RNAP. The structures show that an Ocr dimer binds to RNAP in the cleft, where key regions of sigma bind and where DNA resides during transcription synthesis, thus providing a structural basis for the transcription inhibition. Our results reveal the versatility of Ocr in interfering with host systems and suggest possible strategies that could be exploited in adopting DNA mimicry as a basis for forming novel antibiotics.

*For correspondence:
xiaodong.zhang@imperial.ac.uk

**Competing interests:** The authors declare that no competing interests exist.

## Introduction

Bacteriophage T7 infects *Escherichia coli* and hijacks the host cellular machinery to replicate its genome (*Studier, 1972*; *Krüger and Schroeder, 1981*; *Hausmann and Messerschmid, 1988*). The T7 genome encodes 56 proteins with many functioning as structural proteins for the bacteriophage. A number of T7 proteins are known to specifically inhibit the bacterial cellular machinery. For example, proteins gp0.7, gp2 and gp5.7 inhibit cellular transcription (*Cámara et al., 2010*; *Tabib-Salazar et al., 2018*) whereas gp0.3 inhibits restriction/modification (RM) enzymes (*Studier, 1975*).

Gp0.3 is the first T7 gene expressed after infection and T7 variants lacking gene 0.3 were shown to have genomes susceptible to *E. coli* RM systems (*Studier, 1975*). Subsequently the 117 amino acid protein gp0.3 was named Overcome Classical Restriction (Ocr) (*Krüger and Schroeder, 1981*). Ocr is abundantly expressed and forms a dimer that mimics the structure of a slightly bent 20 base pair B-form DNA (*Issinger and Hausmann, 1972*; *Walkinshaw et al., 2002*) and blocks the DNA binding grooves of the type I RM enzyme, preventing the degradation and modification of the T7 genome by the host. Intriguingly, Type I RM enzymes are present in very low numbers (estimated at ~60 molecules per cell [*Kelleher and Raleigh, 1994*]). Since Ocr is a DNA mimicry protein, it is possible that the abundantly expressed Ocr (estimated to be several hundreds of molecules per cell at least) (*Hausmann and Messerschmid, 1988*) also interferes with other DNA processing systems of the host. Indeed early evidence of an interaction between Ocr and the host RNA polymerase (RNAP) was obtained using pull-down affinity chromatography (*Ratner, 1974*).

**eLife digest** Bacteria and viruses have long been fighting amongst themselves. Bacteriophages are a type of virus that invade bacteria; their name literally means 'bacteria eater'. The bacteriophage T7, for example, infects the common bacteria known as *Escherichia coli*. Once inside, the virus hijacks the bacterium's cellular machinery, using it to replicate its own genetic material and make more copies of the virus so it can spread. At the same time, the bacteria have found ways to try and defend themselves, which in turn has led some bacteriophages to develop countermeasures to overcome those defences.

Many bacteria, for example, have restriction enzymes which recognise certain sections of the bacteriophage DNA and cut it into fragments. However, the T7 bacteriophage has one well-known protein called Ocr which inhibits restriction enzymes. Ocr does this by mimicking DNA, which led Ye et al. to wonder if it could also interrupt other vital processes in a bacterial cell that involve DNA.

Transcription is the first step in a coordinated process that turns the genetic information stored in a cell's DNA into useful proteins. An enzyme called RNA polymerase decodes the DNA sequence into a go-between molecule called messenger RNA, and it was here that Ye et al. thought Ocr might jump in to interfere.

To begin, Ye et al. examined the structure of Ocr when it binds to RNA polymerase using an imaging technique called cryo-electron microscopy. Ocr has been well-studied before, its structure previously described, but not when attached to RNA polymerase. The analysis showed that Ocr gets in the way of specific molecules, called sigma factors, that show RNA polymerase where to start transcription. Ocr binds to RNA polymerase in exactly the same pocket as part of sigma factors do, which is also the place where DNA must be to be decoded to make messenger RNA. Ye et al. then performed experiments to show Ocr interfering with binding to RNA polymerase did indeed disrupt transcription. This means Ocr is quite versatile as it interferes with the RNA polymerase of the bacterial host and its restriction enzymes.

Ocr's strategy of mimicking DNA to interrupt transcription could be adopted as an approach to develop new antibiotics to stop bacterial infections. DNA transcription is an essential cellular process – without it, no cell can replicate and survive – and RNA polymerase is already a validated target for drugs. Following Ocr's lead could provide a new way to stop infections, if the right drug can be designed to fit.

RNA polymerase is the central enzyme for transcription, which is a highly controlled process and can be regulated at numerous distinct functional stages (*Kornberg, 1998*; *Decker and Hinton, 2013*). The large majority of transcription regulation, however, is executed at the recruitment and initiation stage (*Browning and Busby, 2004*; *Hahn and Young, 2011*; *Browning and Busby, 2016*). To ensure transcription specificity, bacterial RNAP relies on sigma (σ) factors to recognise gene-specific promoter regions. *E. coli* has seven sigma factors which can be grouped into two classes, the $\sigma^{70}$ class represented by $\sigma^{70}$, responsible for transcribing housekeeping genes, and the $\sigma^{54}$ class, responsible for transcribing stress-induced genes including phage infection (*Feklístov et al., 2014*; *Browning and Busby, 2016*). Much work has yielded a detailed mechanistic understanding of how transcription directed by $\sigma^{70}$ and $\sigma^{54}$ is initiated (*Zhang et al., 2012*; *Glyde et al., 2018*). Specifically, the two large RNAP subunits β and β' form a crab claw structure that encloses the DNA binding cleft, accommodating the transcription bubble and the downstream double-stranded (ds) DNA (*Bae et al., 2015*; *Zuo and Steitz, 2015*).

Inhibiting host transcription is widely exploited by bacteriophages including T7 on *E. coli* and P23-45 on *T. thermophilus* (*Tagami et al., 2014*; *Tabib-Salazar et al., 2017*; *Ooi et al., 2018*). Gp0.7, gp2 and gp5.7 of T7 were shown to inhibit bacterial RNAP (*Cámara et al., 2010*; *Tabib-Salazar et al., 2017*). Gp0.7 is a protein kinase that phosphorylates the *E. coli* RNAP-$\sigma^{70}$, resulting in transcription termination at sites located between the early and middle genes on the T7 genome, gp2 specifically inhibits $\sigma^{70}$-dependent transcription initiation and gp5.7 inhibits $\sigma^{S}$, responsible for stationary phase adaptation (*Bae et al., 2013*; *Tabib-Salazar et al., 2018*). Two proteins gp39 and gp76 of bacteriophage P23-45 were shown to inhibit *T. thermophilus* transcription (*Tagami et al., 2014*; *Ooi et al., 2018*). Importantly, RNAP is a validated antibacterial target and novel inhibitors of

RNAP hold promise for potential new antibiotic development against antimicrobial resistance (*Ho et al., 2009*; *Srivastava et al., 2011*; *Ma et al., 2016*).

In this work, we investigate the potential effects of Ocr on the host transcriptional machinery. Our results with purified components show that Ocr can bind to RNAP and inhibit both $\sigma^{70}$ and $\sigma^{54}$ dependent transcription. Specifically we establish that an Ocr competes with sigma binding, thus affecting recruitment and can inhibit transcription both in vitro and in *E. coli* cells when over-expressed. Using structural biology, we show that the Ocr dimer directly binds to RNAP at the RNAP cleft, where sigma factors and the transcription bubble bind, suggesting that Ocr interferes with transcription recruitment by competing with sigma factor binding, and could also interfere with transcription initiation. Our work thus reveals the detailed molecular mechanisms of how Ocr could potentially affect the host through inhibiting transcription in addition to its role in knocking out host restriction and modification. These new structural details of RNAP-Ocr complexes allow comparisons with known transcription initiation complexes and other phage proteins suggesting new avenues to be exploited for inhibiting bacterial transcription and combating antibiotic resistance.

## Results

### Ocr interacts with RNAP with high affinity

An early study by *Ratner (1974)* showed that upon T7 infection, Ocr protein was detected in pull-down experiments using host RNAP as bait, suggesting that Ocr could associate with the host transcriptional machinery. To assess if Ocr can engage with RNAP directly, we carried out in vitro interaction studies using purified components (*Figure 1*, *Figure 1—figure supplement 1*). We observed that purified Ocr co-eluted with RNAP in gel filtration experiments, suggesting a tight interaction of RNAP and Ocr sufficient to withstand the lengthy elution time of the complex during chromatography (*Figure 1A–C*). The complex persisted for the order of minutes during chromatography, suggesting a stable complex. To quantify the interactions, we carried out binding experiments between RNAP and Ocr using microscale thermophoresis (MST). Our results show the binding affinity (dissociation constant, Kd) is in the order of 10 nM, similar to that of RNAP with sigma factors (*Figure 1D–F*). These results thus confirm the basis for the RNAP-Ocr interaction originally observed (*Ratner, 1974*). Our observed affinity of Ocr for RNAP is about three orders of magnitude lower than the values reported for the association of Ocr with the EcoKI type I restriction modification enzyme (*Atanasiu et al., 2002*) as would be expected if the type I RM enzyme was the primary target for Ocr.

### Structures of Ocr in complex with RNAP

In order to understand how Ocr interacts with RNAP, we subjected the purified complex from size-exclusion chromatography to cryo electron microscopy (cryoEM) single particle analysis. After several rounds of 2D classification, which allowed us to remove ice contaminations, very large particles or particles much smaller than RNAP, the remaining particles were subject to 3D classifications using the structure of the RNAP filtered to 60 Å as an initial model (PDB code 6GH5, *Figure 2—figure supplement 1*). Two classes with clear density for Ocr were refined to a global resolution 3.7 and 3.8 Å respectively although the quality is not uniform in the reconstruction, as demonstrated in the resolution distribution. Two resolution zones seem to be evident, around 3–4 Å and around 7–8 Å, reflecting the structural features of the complex (some domains are mobile) and partly due to preferential orientations of the particles (*Figure 2*, *Figure 2—figure supplement 2* and *Figure 2—figure supplement 3*). The electron density for RNAP is clearly resolved and two distinct structural models of RNAP (taken from PDB code 6GH6 and 6GH5) can be readily fitted into the two reconstructions (*Figure 2* and *Figure 2—figure supplement 3A*). The remaining density region that is not accounted for by RNAP can accommodate an Ocr dimer (*Figure 2—figure supplement 3A*). The density for Ocr is of sufficient quality that allows accurate positioning of the Ocr atomic structure (PDB code 1S7Z) (*Figure 2—figure supplement 3B–3C*). The regions with the highest resolution within the reconstruction are in the RNAP core (*Figure 2—figure supplement 2*), where density for many side chains is clearly visible (*Figure 2—figure supplement 3C*).

During 3D classification, it is clear that a number of different conformational and structural classes exist within the dataset and two distinct classes were identified due to their widely differing RNAP

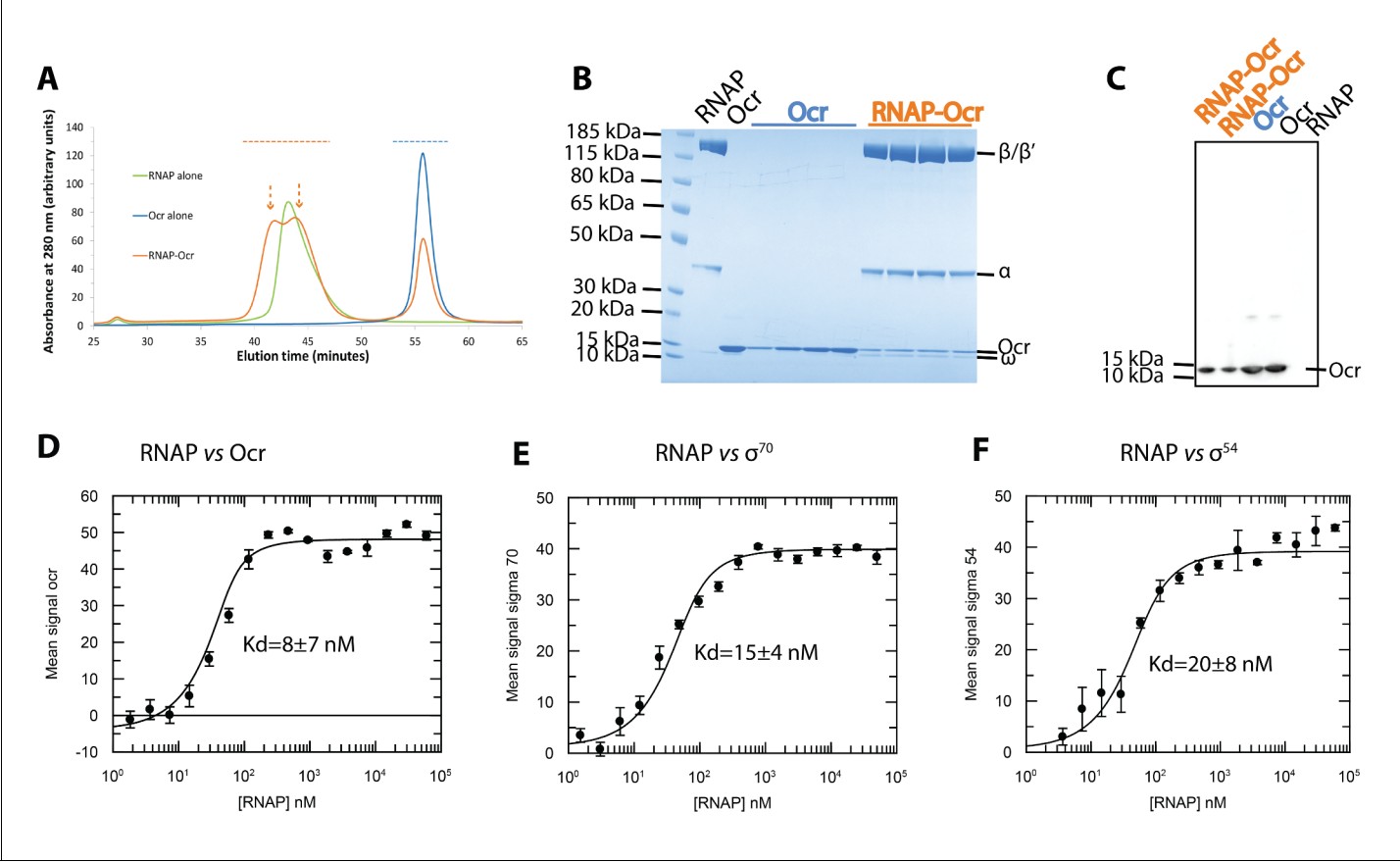

**Figure 1.** RNAP interacts with Ocr and can form a stable complex with Ocr. (A) Gel filtration chromatography profiles of RNAP (green), Ocr (blue) and RNAP-Ocr complex (orange). Two peaks correspond to RNAP-Ocr complexes (B) SDS-PAGE of the corresponding fractions from (A) as indicated by coloured dashed lines in (A) and the purified RNAP and Ocr as comparison, the same below in (C). (C) Western blotting of samples from RNAP-Ocr gel filtration fractions to verify the presence of Ocr co-eluting with RNAP, here only Ocr is his-tagged. Two fractions under each peak (as coloured dash arrows shows) in gel-filtration is loaded. (D–F). MST experiments measure the dissociation constants (Kd) of Ocr, $\sigma^{70}$ and $\sigma^{54}$ with RNAP. These Kd values represent maximum values.

The online version of this article includes the following figure supplement(s) for figure 1:

**Figure supplement 1.** Protein purification and complex formation.

clamp conformations (*Figure 2—figure supplement 1*). Bacterial RNAP consists of five subunits forming a crab claw shape with the two large subunits β and β' forming the claws (or clamp) which enclose the RNAP channel that accommodates the transcription bubble and downstream double-stranded (ds) DNA. The RNAP clamp is highly dynamic and single-molecule Forster Resonant Energy Transfer data have shown that the RNAP clamp adapts a range of conformations from closed to open involving more than 20 Å movement and 20° rotation of the clamp (*Chakraborty et al., 2012*). Our recent structural work showed that specific clamp conformations are associated with distinct RNAP functional states with the widely opened clamp conformation associated with the DNA loading intermediate state (*Glyde et al., 2018*).

In both of the distinct RNAP-Ocr complex structures, Ocr remains as a dimer upon binding to RNAP (*Figure 2A–B*) and the Ocr dimers are slightly less bent compared to that of an Ocr alone crystal structure (*Figure 2—figure supplement 4*). In one structure, the entire Ocr dimer inserts deeply into the RNAP cleft, occupying the full RNAP nucleic acid binding channel including where the transcription bubble and the downstream DNA resides in the transcriptional open and transcribing complex (*Figure 2A*, *Figure 2C*, *Figure 3A*). In this interaction mode, the RNAP clamp is wide open and both Ocr subunits are involved in the interactions (*Figure 2A*, *Figure 3B*, *Figure 2—figure supplement 3*), we thus denote it as the 'wide clamp structure'. The second structure involves only

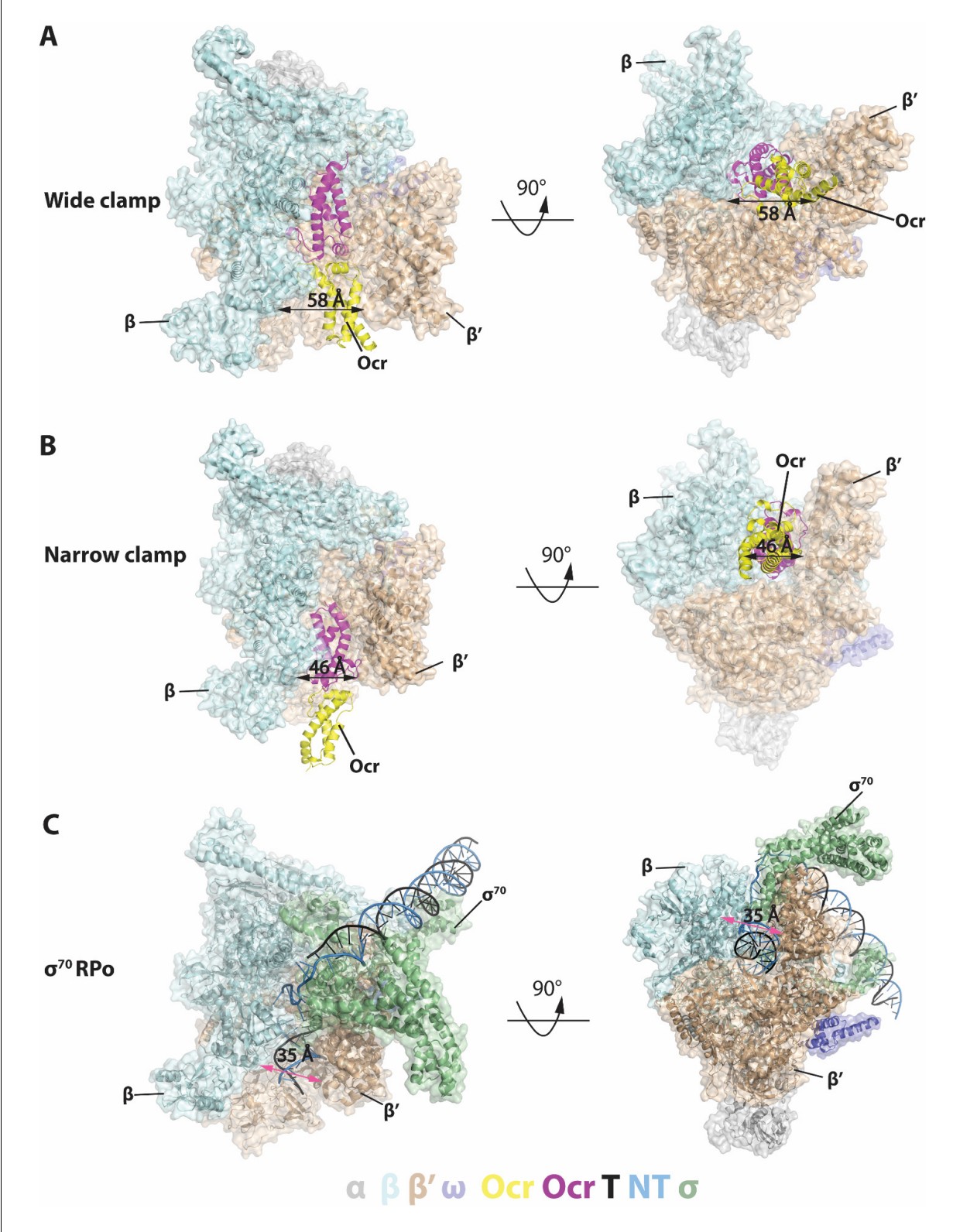

**Figure 2.** Structures of RNAP-Ocr in two different binding modes in two orthogonal views. (A) 'wide-clamp' mode. (B) 'narrow-clamp' mode. RNAP and $\sigma^{70}$ were shown as surface, Ocr shows as cartoon. The colour key is below the figure. α-grey, β-pale cyan, β'-wheat, ω-slate, $\sigma^{70}$-palegreen, Ocr–magenta (proximal subunit) and yellow (distal subunit), non-template strand DNA(NT)-sky blue, template strand DNA (T)-black. (C) RNAP-$\sigma^{70}$ open promoter complex (PDB code 6CA0).

*Figure 2 continued on next page*

*Figure 2 continued*

The online version of this article includes the following figure supplement(s) for figure 2:

**Figure supplement 1.** cryoEM data quality and Image processing flowchart.
**Figure supplement 2.** Quality of the two distinct RNAP-Ocr complex 3D reconstructions.
**Figure supplement 3.** Representative electron density maps in the two reconstructions.
**Figure supplement 4.** Structural comparison of Ocr from RNAP-Ocr complex (magenta and yellow) with that from crystal structure (blue, PDB code 1S7Z).

one of the two Ocr subunits within the dimer contacting RNAP, occupying only the downstream DNA channel. Here the RNAP clamp is less open and we denote this structure the 'narrow clamp structure' (*Figure 2B*, *Figure 2—figure supplement 3*).

In the wide clamp structure, the RNAP clamp conformation is very similar to the conformation observed in the transcription initiation intermediate complex where the clamp is wide open and DNA is partially loaded into the cleft *Glyde et al., 2018*). In this structure, the Ocr dimer inserts into the RNAP channel and each of the two Ocr subunits, which we term proximal and distal, and makes distinct interactions with RNAP. The proximal subunit is deeply embedded in the RNAP cleft and occupies the space for the template strand DNA, the complementary newly synthesised RNA and the non-template strand DNA (*Figure 2* and *Figure 3*, magenta [*Bae et al., 2015*; *Zuo and Steitz, 2015*; *Glyde et al., 2018*]). The distal Ocr subunit occupies the position of the downstream DNA (*Figure 2* and *Figure 3A*, yellow). The negatively charged surface of the proximal Ocr complements the highly positively charged surface of the surrounding RNAP and fits snugly into the channel

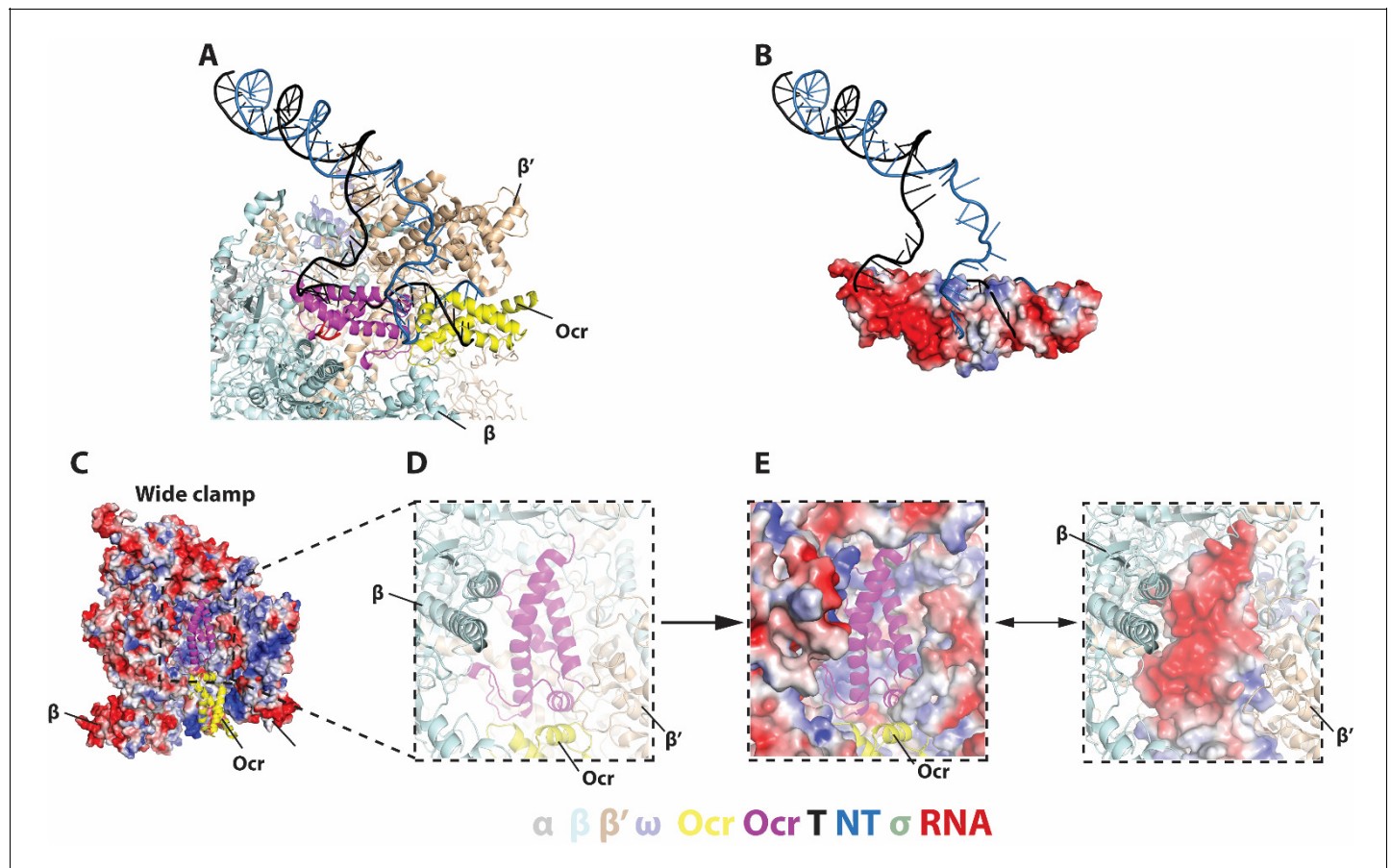

**Figure 3.** Detailed interactions between Ocr and RNAP and in the 'wide-clamp' conformation. (**A–B**) Comparison with initial transcribing complex structure (PDB 4YLN). (**C**) Ocr resides in the positively charged RNAP channel. (**D–E**) Detailed charge distributions of RNAP and Ocr in the interacting regions. Blue – positive charge, red – negative charge.

formed by β and β' subunits (*Figure 3C–E*). The distal Ocr subunit mimics the downstream DNA in its interactions with the β' clamp. Superposition of the β' clamp in the RNAP-Ocr structure with that of initial transcribing complex shows the distal Ocr monomer aligns with downstream dsDNA in the initial transcribing complex (*Figure 3B*). However, the Ocr dimer has a rigid conformation that resembles a bent B-DNA. Consequently, in order to maintain the interactions between both Ocr subunits and RNAP, the β' clamp has to open up (to ~58 Å in RNAP-Ocr from ~35 Å in RPo) to accommodate the Ocr dimer compared to RPo that accommodates a transcriptional bubble and dsDNA (*Figure 2*).

In the narrow clamp mode, the proximal Ocr subunit interacts with the β' clamp, again mimicking downstream DNA in RPo although the clamp is more open (~45 Å) compared to that in RPo (~35 Å) (*Figure 2B–C*). Interestingly, although an Ocr monomer surface mimics that of a dsDNA in the overall dimension and the negative charge distributions, the Ocr subunit in the narrow clamp structure does not exactly overlay with the downstream dsDNA in RPo when RNAP is aligned on its bridge helix, the highly conserved structural feature that is close to the active centre and connects β and β' clamps (*Figure 4A*). Instead Ocr is shifted upwards compared to the dsDNA in RPo (*Figure 4A*). Inspecting the structures of Ocr and RNAP explain the differences. The downstream binding channel of RNAP consists of β and β' clamps on two sides and the β' jaw domain as a base, providing a positively charged environment on three sides, required to accommodate and engage with the negatively charged DNA during transcription (*Figure 4B*). Although an Ocr monomer has a largely negatively charged surface, there are also positively charged patches on the Ocr surface, especially those facing the β' jaw domain (*Figure 4C*). In order to maintain the interactions between Ocr and the β' clamp and to overcome charge repulsions with the β' jaw, the β' clamp is opened up, shifting Ocr upwards and away from the jaw domain.

## Ocr inhibits transcription in vitro and in cells

In order to understand the significance of Ocr binding to RNAP, we investigated the effects of Ocr on in vitro transcription using short-primed RNA (spRNA) assays performed on various promoter DNAs (see Materials and methods). To determine the effect of Ocr on RNAP-$\sigma^{70}$ transcription activity we performed in vitro transcription assays using different promoters and promoter variants such as linear $\sigma^{70}$ promoter, or *lac*UV5 promoter (*Cámara et al., 2010*) as well as a T7A1 supercoiled promoter (*Figure 5A–B*, *Figure 5—figure supplement 1*). Our data reveal that adding 5 µM Ocr decreased activity of RNAP-$\sigma^{70}$ on the different promoters in the range of 70–10%. On the linear promoter in the presence of 5 µM Ocr the amount of spRNA was significantly reduced (to 10–30%) when Ocr was added prior to or after the holoenzyme formation but before DNA was added (*Figure 5A-B* I-II). However, the effect of Ocr after DNA was added was less pronounced (*Figure 5A–B III*). Ocr did not abolish transcription on the supercoiled T7A1 promoter at the same extent as on the linear promoters, indicating a role for DNA topology and perhaps the energy barriers of DNA opening in open complex formation and the promoter sensitivity to Ocr (*Figure 5—figure supplement 1C* I-III).

The effect of Ocr on the major variant RNAP-$\sigma^{54}$ dependent transcription was also assessed using different variants of *Sinorhizobium meliloti nifH* promoter including linear and supercoiled DNA (*Chaney and Buck, 1999*; *Glyde et al., 2018*), (*Figure 5—figure supplement 1A,B*). The phage shock protein F (PspF$_{1-275}$) was used as an activator of RNAP-$\sigma^{54}$ transcription to promote open complex formation (see Materials and methods). Again, on both linear and supercoiled *nifH* promoters the RNAP-$\sigma^{54}$ complexes were sensitive to Ocr (*Figure 5—figure supplement 1*).

Our results show that Ocr can efficiently inhibit transcription in vitro. We next tested to see if expression of Ocr in cells could indeed exhibit an effect in cells. We used *E. coli* MG1655 cells expressing Ocr from a plasmid vector under the control of an arabinose-inducible promoter and a range of growth conditions (*Supplementary file 1*).

An Ocr-mediated growth inhibition effect was indeed observed in minimal modified M9 media (*Figure 5C*). The effects were further assessed using the chromosomal β-galactosidase gene as a reporter. During the 5 hr period following induction with IPTG, a ~ 10 fold decrease of normalised β-galactosidase activity was observed in *E. coli* cells expressing Ocr compared to cells that do not express Ocr (*Figure 5D*). Moreover, at 15 hr post-induction, upon induction, there was a 4-fold increase of normalised β-galactosidase activity in cells not expressing Ocr, while the β-galactosidase activity in cells expressing Ocr was lower overall and the observed increase upon induction was less

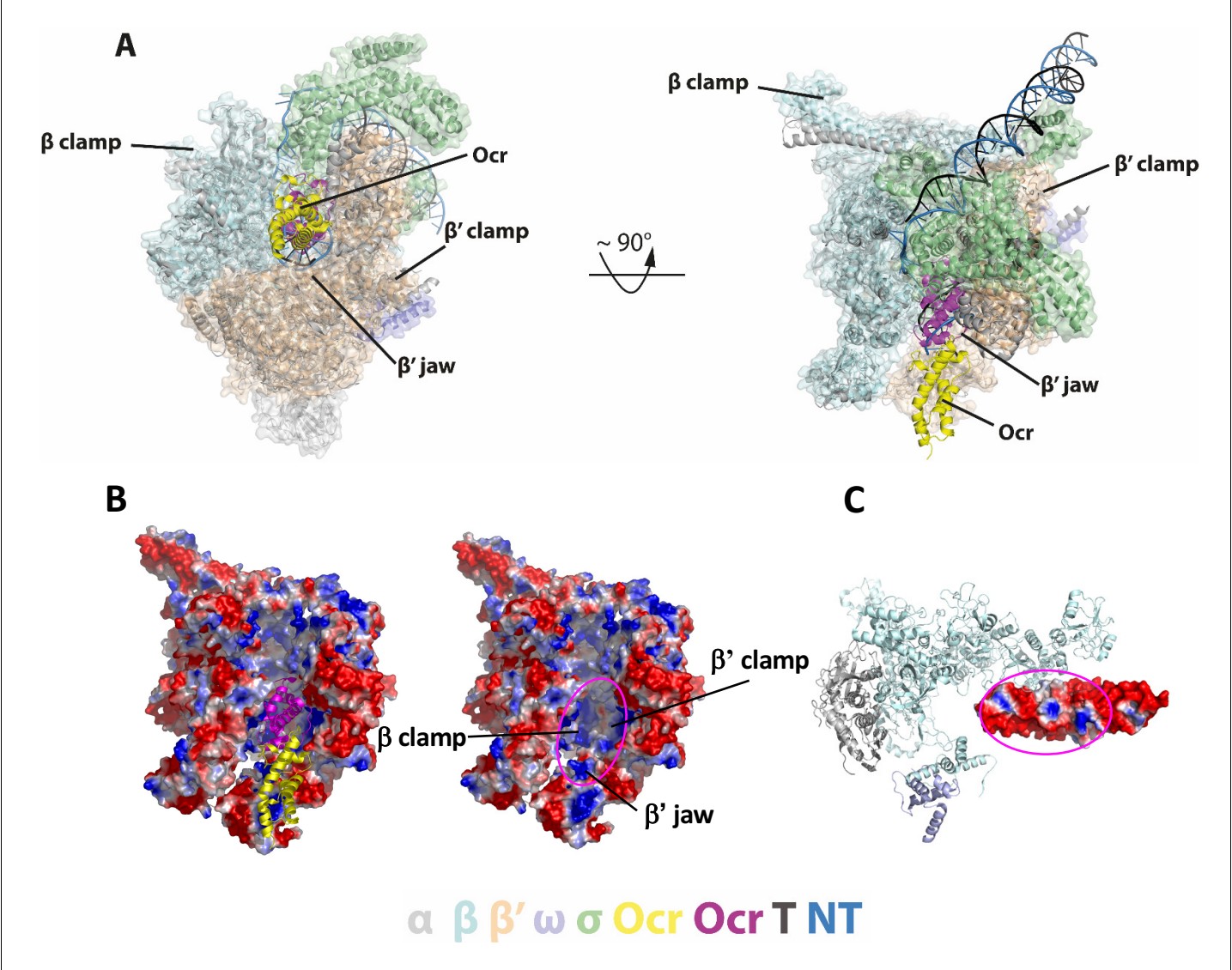

**Figure 4.** Detailed interactions between RNAP and Ocr in the 'narrow-clamp' conformation. (**A**) Two orthogonal views of RNAP-Ocr narrow-clamp structure (RNAP shows as surface and Ocr shows as cartoon (magenta and yellow), overlaid with σ70 open complex structure (PDB code 6CA0). (**B**) Surface charge distribution of RNAP showing positively charged surfaces in the cleft where Ocr binds. Magenta ellipse – where Ocr binds. (**C**) Surface charge distribution of Ocr facing the bottom of the cleft including β' jaw domain. Magenta ellipse – area of Ocr that is inside the cleft showing both negative and positively charged areas. Blue – positively charged, red – negatively charged.

than 2-fold (*Figure 5—figure supplement 2A*). Subsequent measurements of *lacZ* mRNA using reverse transcription followed by quantitative polymerase chain reaction (RT-qPCR) showed that the observed β-galactosidase activity is concomitant with *lacZ* mRNA levels (*Figure 5—figure supplement 2B*), confirming that Ocr was inhibitory for transcription *in cells*.

## Ocr inhibits RNAP recruitment and open complex formation but does not disrupt pre-formed open complexes

Since Ocr interfered with transcription, we wanted to investigate whether it does so through interfering with RNAP recruitment by sigma and/or other processes required for transcription. Specifically, we probed the ability of Ocr in interfering with RNAP- σ holoenzyme and holoenzyme-promoter complex formation (recruitment) and pre-formed open complex, which is competent for transcription, thus allowing us to assess the effects on elongation.

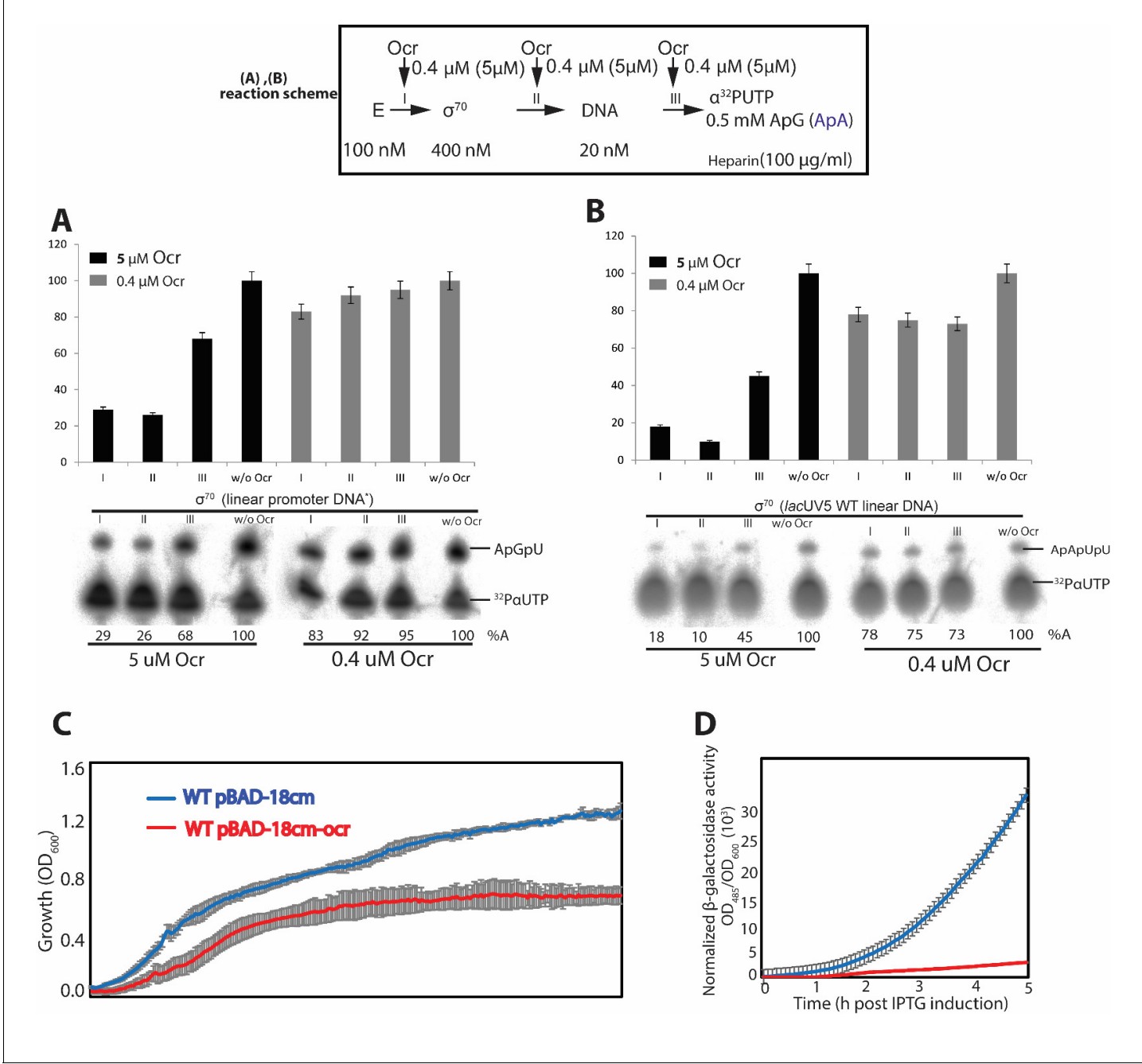

**Figure 5.** In vitro transcription and in vivo reporter assays. (A-B) spRNA experiments on σ⁷⁰ using two different promoter DNAs. Reaction schematics are shown with I, II, III representing experiments when Ocr is added during different stages of transcription initiation reaction. A control lane without Ocr is also shown. (C-D) using a reporter assay, we show that over-express of Ocr in *E. coli* cells inhibits cell growth (C) and β-galactosidase activities (D). At least three independent experiments were performed, each with 3–5 technical replicates.

The online version of this article includes the following figure supplement(s) for figure 5:

**Figure supplement 1.** In vitro transcription assays of σ⁵⁴-dependent systems using linear (A) and supercoiled DNA (B) and σ⁷⁰ transcription using super-coiled DNA (C).

**Figure supplement 2.** in vivo data showing Ocr can inhibit transcription.

Using purified components and gel mobility shift assays, Ocr was observed to inhibit RNAP-$\sigma^{70}$ holoenzyme formation in a concentration dependent fashion and irrespective of whether Ocr is added before or after σ (*Figure 6A and B*, comparing lane 6 with 7–10). Direct competition experiments with pre-formed RNAP-$\sigma^{70}$ holoenzyme show that appearance of a RNAP-Ocr complex accompanies the disappearance of the RNAP-$\sigma^{70}$ holoenzyme, (*Figure 6A and B*, comparing lane 6 with lanes 11–12). Indeed, MST experiment of Ocr binding to RNAP-$\sigma^{70}$ revealed that Ocr can displace $\sigma^{70}$ with an apparent Kd of ~20 nM (*Figure 6—figure supplement 1C*). This is also consistent with data showing that RNAP has similar affinities for Ocr and σ (*Figure 1D–F*).

We next assessed the ability of Ocr in inhibiting recruitment to the promoter DNA by monitoring the amount of DNA bound holoenzymes using two different promoters (*Figure 6C–D*). Ocr clearly inhibits the formation of a RNAP-$\sigma^{70}$-DNA complex in a concentration dependent manner. Adding 5 µM Ocr either before or after σ addition resulted in ~30% promoter complex formation (*Figure 6C–D,I–II*). Adding Ocr after DNA has only limited effects (*Figure 6C–D III*). This is consistent with the higher binding affinity between RNAP-σ and DNA (0.1 nM for $\sigma^{70}$ and 3 nM $\sigma^{54}$) (*Figure 6—figure supplement 1*) compared to RNAP-Ocr (*Figure 1D*). These results are in excellent agreement with those in vitro transcription assay (*Figure 5*).

Similar effects were observed for RNAP-$\sigma^{54}$ (*Figure 6—figure supplement 1*). Due to the lower sensitivity of the diffused $\sigma^{54}$ band to Coomassie blue staining, the displaced free $\sigma^{54}$ from the RNAP-$\sigma^{54}$ holoenzyme was observed by western blotting with an antibody to the His-tag on $\sigma^{54}$.

Together our binding experiments and in vitro transcription assays support the proposal that Ocr inhibits RNAP recruitment to the promoter DNA by directly competing with σ for RNAP binding. However, Ocr has only limited effects on pre-formed RNAP-σ-DNA complexes. This is similar to the interactions of Ocr with a type I RM enzyme where Ocr only affects the enzyme when not bound to DNA (*Bandyopadhyay et al., 1985*).

## Discussion

### Ocr is a bifunctional protein

Our data here characterise a new activity for Ocr in inhibiting the host transcriptional machinery in addition to its established role in inhibiting the type I RM systems of the host. Thus, Ocr is a bifunctional DNA mimic protein.

The in vitro transcription assay and Ocr competition experiments clearly show that Ocr can inhibit σ binding and recruitment of RNAP to promoter DNA. Our data using a reporter assay also demonstrate the ability of Ocr in inhibiting in vivo the *E. coli* transcription machinery. The structures of Ocr-RNAP show that Ocr occupies the space where σ binds (*Figure 7A–B*) and where the transcription bubble and downstream DNA reside in the open complex and elongation complex, thus explaining how Ocr inhibits RNAP recruitment by σ (*Vassylyev et al., 2007*; *Zuo and Steitz, 2015*).

Our data suggest an answer to the puzzle of why Ocr is so abundantly overexpressed immediately upon a T7 infection (*Issinger and Hausmann, 1972*; *Studier, 1975*), when its high affinity classical target, the Type I RM enzyme, is present in very low numbers (estimated at ~60 molecules per cell, [*Kelleher and Raleigh, 1994*]). The very tight binding between Ocr and Type I RM enzymes ensures complete inhibition of restriction and modification. It would appear from our data that the excess Ocr molecules could then start the process of shutting down host transcription by binding to any non-transcribing RNAP. An earlier study by McAllister and Barrett used host strains lacking the host restriction systems, thus eliminating the effects of Ocr on host restriction enzymes. They showed that expression of gp0.3 (Ocr) resulted in a delayed inhibition of host β-galactosidase synthesis and delayed inhibition of total RNA synthesis (*McAllister and Barrett, 1977*). This inhibitory effect is consistent with our studies here showing that Ocr can inhibit host transcription. Since Ocr is expressed by host RNAP, it thus takes time for a sufficient amount of Ocr to be synthesized by the host, resulting in the delayed response. Indeed, Ocr could bind to free RNAP and/or potentially compete with host $\sigma^{70}$ which are in the range of hundreds of molecules per cell (https://pax-db.org/dataset/511145/137/), since their binding affinities to RNAP are similar (*Figure 1D–F*) and Ocr can compete with $\sigma^{70}$ for RNAP binding and promoter DNA binding (*Figure 6—figure supplement 1C*). Interestingly, when Ocr and gp1.1 (a small protein with unknown function) were both expressed, the

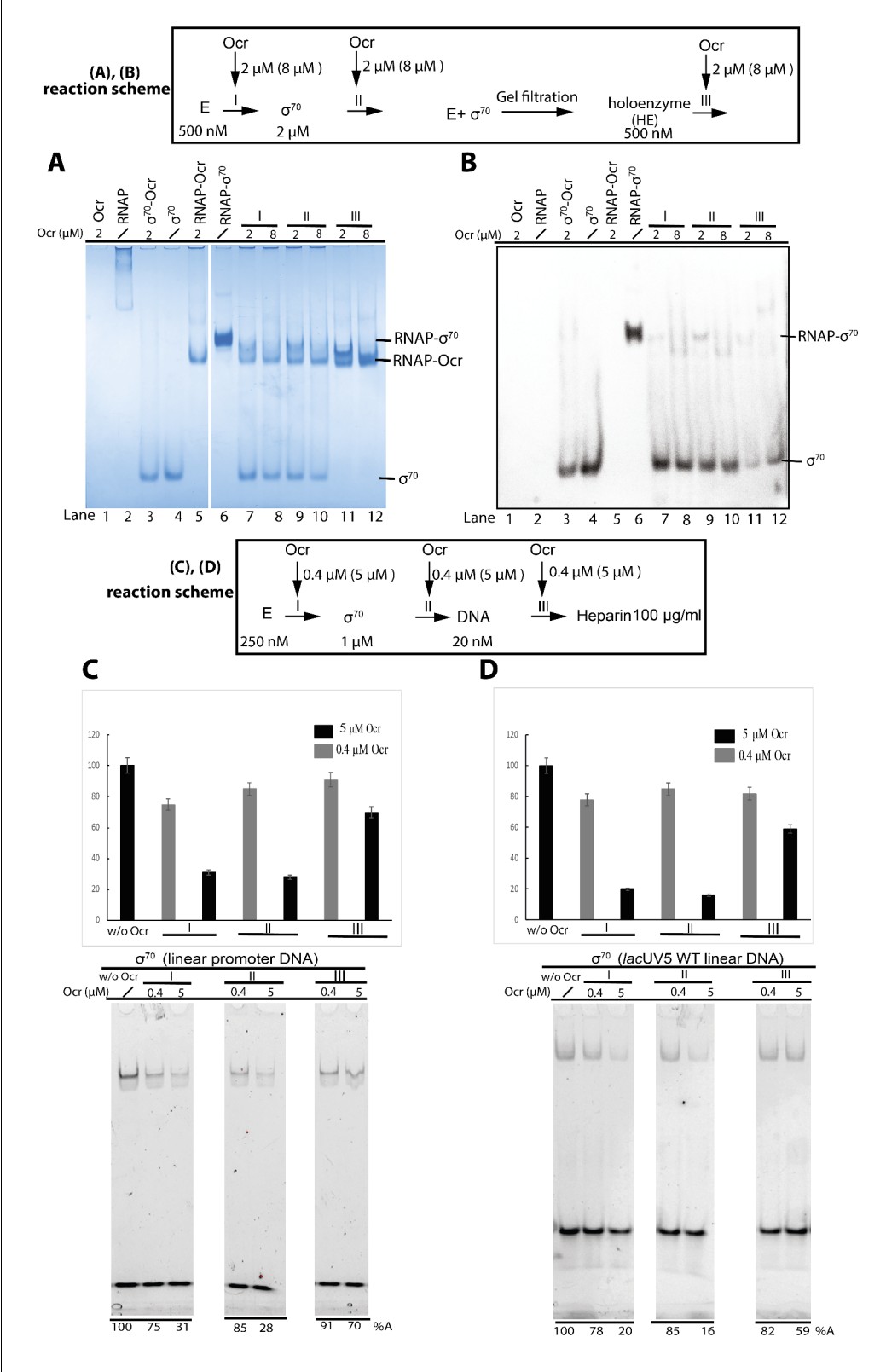

**Figure 6.** Ocr and its effect on σ70 holoenzyme and open complex formation. (A and B) Native PAGE shows Ocr can inhibit and disrupt RNAP-σ70 holoenzyme formation. Coomassie blue staining (A) and Western blots against His-tagged σ70 (B). (A and B) reaction schemes are indicated above. E represents RNAP core enzyme; I, II, III indicate the point when Ocr was added during the reactions. In Reaction III, holoenzyme was preformed and purified by size exclusion chromatography first. Protein concentrations are shown in the reaction schemes. In lanes 7, 9, 11, a 1:1 molar ratio of Ocr to σ

*Figure 6 continued on next page*

*Figure 6 continued*

was used, whereas in lane 8, 10, 12, a 4:1 molar ratio of Ocr to σ was used. The exact Ocr concentrations are labelled above each lane. (C and D) $\sigma^{70}$ open complex formation with different promoter DNA as assayed by native-PAGE gels and stained for DNA. Reaction schemes are indicated above. I, II, III indicate the point when Ocr was added during the reactions. The same experimental conditions were used as in vitro transcription assays in *Figure 5*, with either 0.4 µM or 5 µM final concentration Ocr added for each reaction, as indicated in each lane. Reaction in the absence of Ocr (w/o Ocr) was used as a control. All reactions include 100 µg/ml heparin to ensure only open complexes are captured.

The online version of this article includes the following figure supplement(s) for figure 6:

**Figure supplement 1.** Ocr interferes with holoenzyme formation of $\sigma^{54}$(A-B).

host protein synthesis was shutdown effectively, suggesting that gp1.1 and Ocr might have complementary/synergetic effects.

The bifunctionality observed in Ocr may also be shared by other DNA mimic proteins. For example, the phage T4 Arn protein targeting the McrBC restriction enzyme also has weak interaction with histone-like protein H-NS (*Ho et al., 2014*). Arn is very similar in shape to Ocr but since T4 requires

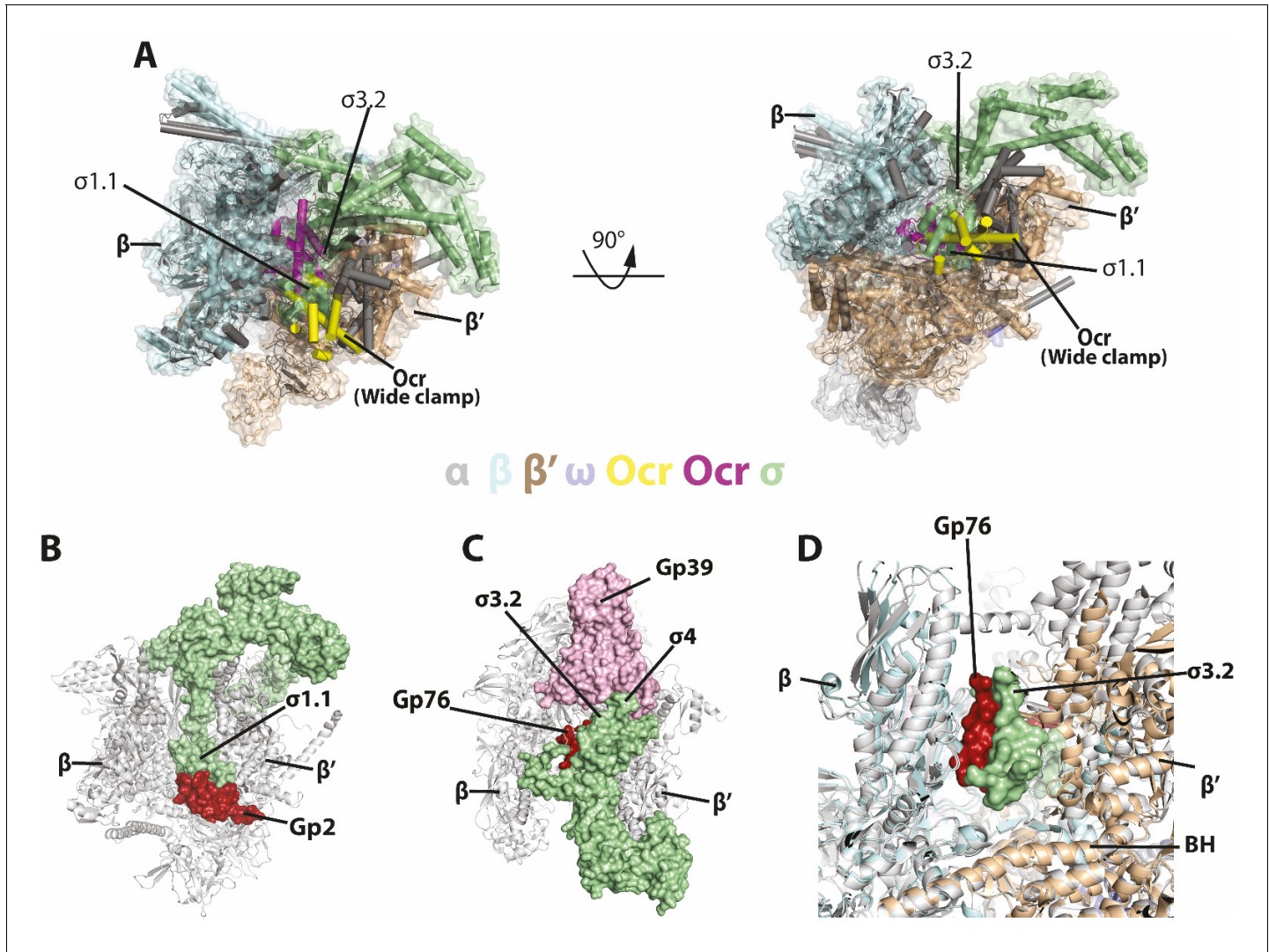

**Figure 7.** Comparisons with sigma and other phage proteins in binding to RNAP. (A) Ocr overlay with RNAP-$\sigma^{70}$ holoenzyme B) Complex structure of gp2 of T7 with $\sigma^{70}$ holoenzyme (PDB 4LLG), (C) Complex structure of gp76-gp39 of P23-45 with $\sigma^{70}$ holoenzyme (PDB 5XJ0), (D) Enlarged view of gp76 and $\sigma^{70}$. RNAP shows as cartoon and coloured as grey, $\sigma^{70}$ shows as surface and coloured pale green. Gp2, gp39, gp76 are show as surface and coloured as firebrick, light pink and firebrick, respectively.

the host RNAP throughout its life cycle it is unlikely to target RNAP as strongly as Ocr. Additional functions may also exist for other DNA mimics like phage λ Gam protein (*Wilkinson et al., 2016*) and the recently discovered anti-CRISPR proteins (*Wang et al., 2018*). How the multifunctionality of these proteins are utilised by phage is an interesting topic for further study.

## Comparisons with other RNAP inhibiting bacteriophage proteins

Several other bacteriophage proteins inhibit transcription. These include T7 gp2 and gp5.7, which inhibits *E. coli* RNAP, and P23-45 gp39 and gp76, which inhibit *T. thermophilus* transcription. Structural and biochemical studies show that gp2 specifically inhibits $\sigma^{70}$-dependent transcription through the interactions and stabilisation of the position of the inhibitory region 1.1 of $\sigma^{70}$, between the β and β' clamps at the rim of the downstream DNA binding channel (*Figure 7B*; *Bae et al., 2013*). Gp39 binds at region 4 of $\sigma^{A}$ and restricts its conformation so that it is unable to bind to −35 promoter regions (*Figure 7C*; *Tagami et al., 2014*). Gp76 on the other hand, binds deeply in the RNAP cleft, preventing open complex formation as well as transcription elongation (*Figure 7D*; *Ooi et al., 2018*). However, the binding site of gp76 is adjacent to, but not overlapping the $\sigma^{70}$ region $\sigma_{3.2}$, thus gp76 does not inhibit holoenzyme formation (*Figure 7D*).

In all the cases above, the bacteriophage proteins do not inhibit holoenzyme formation. Ocr on the other hand, inhibits holoenzyme formation per se and can disrupt a pre-formed holoenzyme (*Figure 6A–B*, *Figure 6—figure supplement 1*, *Figure 7A*). In addition, Ocr can also inhibit open complex formation but is unable to effectively disrupt pre-formed open complex (*Figure 6*).

## Unique features of Ocr defines its ability in specific DNA processing systems

Ocr was shown to mimic a dsDNA in both shape and charge distribution (*Walkinshaw et al., 2002*) and an Ocr dimer mimics a slightly bent B-form DNA. Ocr has been shown to occupy the DNA binding grooves of type I RM enzymes through specific interactions, thus preventing viral DNA from being degraded or modified by the host RM system (*Atanasiu et al., 2002*; *Kennaway et al., 2009*).

The structural model of the archetypal Type I RM enzyme EcoKI (*Kennaway et al., 2009*) shows a close association of Ocr with the M.EcoKI modification methyltransferase core of the Type I RM enzyme in that Ocr is completely encircled by the enzyme, similar to DNA encircled by a DNA clamp (*Kennaway et al., 2012*; *Walkinshaw et al., 2002*) (PDB 1S7Z). Ocr, on the other hand, binds in the cleft, surrounded on three sides by RNAP. Therefore it is not surprising that both Ocr-RNAP and Ocr-M.EcoKI complexes are sufficiently stable to persist during lengthy size exclusion chromatography experiments (*Figure 1* and *Atanasiu et al., 2002*).

In this work, we show that Ocr remains as a dimer and binds at the downstream DNA channel of RNAP via the complementary charge interactions between the positively charged RNAP channel and the negatively charged Ocr dimer. However, due to the extensive positively charged surface within RNAP, the binding is not specific. Indeed, two distinct binding modes have been detected which allow either the proximal or distal Ocr monomer to interact with the positively charged β' clamp. The RNAP clamp opens up in the 'wide clamp' binding mode in order to accommodate the Ocr dimer conformation while maintaining the charge complementarity, both deep inside the cleft and with the β' clamp. When the clamp is closed as in the transcriptional open complex, even though there is ample space to accommodate one Ocr subunit, the charge repulsion between the β' jaw domain and the Ocr surface results in the opening of the β' clamp, thus shifting Ocr upwards and away from the β' jaw domain. These structures demonstrate that the structural flexibility of RNAP clamp plays a key role in its ability to accommodate Ocr. A study of the kinetics of the formation of the complexes between Ocr and RNAP would be interesting and may show multiple consecutive reactions as found for the interaction of a small DNA mimic Uracil glycosylase inhibitor (UGI) with uracil glycosylase (*Bennett et al., 1993*).

The interactions and structures of Ocr with RNAP observed here suggest that Ocr could potentially interact and inhibit other DNA processing enzymes through a series of non-specific interactions with dsDNA binding sites/channels. However, the rigidity and the bent conformation of the Ocr dimer imposes constraints on the binding sites, which are optimised for the RM enzymes it inhibits but can still be accommodated by the structurally flexible RNAP DNA binding channel. The rigidity

of the dimer is defined by the interactions between the Ocr subunits. Work presented here suggests that Ocr could be modified to fine-tune its DNA mimicry for binding to specific DNA processing proteins, either by reducing the rigidity of the dimer or changing the bend angle to create a new dimer interface. DNA mimicry has been proposed as a potential effective therapeutic tool (*Putnam and Tainer, 2005*; *Dryden, 2006*; *Roberts et al., 2012*), Ocr could thus be exploited to create specific variants that can target specific DNA binding proteins, especially those that are shown to be antibiotic targets. In addition, our data here show that Ocr can effectively inhibit transcription under stress conditions when transcriptional activity is reduced. It is thus possible that during other growth conditions/phases, such as stationary phases or persistence, Ocr could be effectively utilised to inhibit transcription and thus diminish bacterial survival.

## Materials and methods

### RNAP-Ocr complex formation

*E. coli* RNA polymerase was purified as previously reported (*Yang et al., 2015*). As well as being purified as previously described (*Sturrock et al., 2001*), Ocr was ligated into pOPINF vector with a N-terminal His-tag. Three purification steps (Ni-NTA affinity, Heparin, and Gel filtration chromatography) were employed, which generated homogeneous Ocr protein as judged by SDS-PAGE gel. The RNAP-Ocr complex was purified by gel filtration chromatography. Initially RNAP (final concentration 14.5 µM) and Ocr (all Ocr concentrations are for the dimeric form, final concentration 58.0 µM) were mixed at a 1:4 molar ratio in a 150 µl reaction volume and the mixture (150 µl) was incubated on ice for 30 mins before being loaded onto the gel filtration column with a flowrate of 0.3 ml/min (Superose 6 10/300, GE Healthcare) and eluted in buffer containing 20 mM Tris pH8.0, 150 mM NaCl, 1 mM TCEP, 5 mM MgCl$_2$. The fractions were run on SDS-PAGE and the presence of Ocr was confirmed by Western blot against the His-tag. The same amount and concentration of RNAP and Ocr alone were also loaded on a Superose 6 10/300 column separately as a control .

### Microscale thermophoresis (MST)

All MST experiments were performed using a Monolith NT.115 instrument (NanoTemper Technologies, Germany) at 22 ˚C. PBS (137 mM NaCl, 2.7 mM KCl, 10 mM Na$_2$HPO$_4$, 1.8 mM KH$_2$PO$_4$) supplemented with 0.05% Tween-20 was used as MST-binding buffer for all experiments. In all cases hydrophobic treated capillaries were used. $\sigma^{54}$, $\sigma^{54}$-R336A and $\sigma^{70}$ were purified to homogeneity similarly as described previously (*Nechaev and Severinov, 1999*; *Yang et al., 2015*). $\sigma^{54}$-R336A and $\sigma^{70}$ holoenyzme were individually produced with gel filtration chromatography by mixing RNAP and σ with 1:4 molar ratio and incubating on ice for half an hour before loading to superpose 6 10/300 column.

Ocr, $\sigma^{54}$, and $\sigma^{70}$, $\sigma^{54}$-R336A -RNAP holoenzyme and $\sigma^{70}$-RNAP holoenzyme were all His tagged and labelled with the kit (Monolith His-Tag Labeling Kit RED-tris-NTA 2nd Generation) separately. The labelled Ocr and sigma factors were diluted to a concentration of 50 nM and mixed with an equal volume of a serial dilution series of RNAP incubated at room temperature for 20 min before loading into MST capillaries thus giving a final concentration of 25 nM for Ocr and the sigma factors in the direct titration of RNAP. For Ocr competitive binding with $\sigma^{70}$ holoenzyme, 50 nM of his-labeled $\sigma^{70}$ holoenzyme, diluted in MST-binding buffer, was mixed with equal volumes of a dilution series of Ocr in MST-binding buffer. For DNA with holoenzyme binding affinity, linear promoter DNA (*Zuo and Steitz, 2015*) was used $\sigma^{70}$ and a *nifH* promoter with mismatch DNA between -10 and -1 (*Glyde et al., 2018*) was used. 40 nm of his-tagged holoenzyme was mixed with equal volumes of a dilution series of DNA in MST-binding buffer. MST experiments were performed using 70–100% LED power and 40–60% MST power with a wait time of 5 s, laser on time of 30 s and a back-diffusion time of 5 s. For all the binding experiments (Ocr or σ with RNAP, or DNA with holoenzyme), MST data were analysed using Grafit 3.0 (Erithacus Software) and the data were fitted with a quadratic tight binding equation with the concentration of the constituent being titrated kept constant in the fitting process. The dissociation constants obtained are maximum values and the real values are likely to be rather smaller. This is due to the concentration of the protein being titrated being larger than the dissociation constant. In the competitive binding experiment with Ocr and $\sigma^{70}$-

RNAP holoenzyme, the data were fitted using the tight binding equation to determine the apparent dissociation constant. Each experiment was repeated at least three times.

## CryoEM sample preparation

For structural studies using cryoEM, the His-tag of Ocr was cleaved and the purification procedure was as previously reported (*Sturrock et al., 2001*). The complex was formed as described above. The RNAP-Ocr complex sample was concentrated to 0.6 mg/ml and 3.5 µl of the complex was applied to R2/2 holey carbon grids (Quantifoil). The vitrification process was performed using a Vitrobot Mark IV (FEI) at 4°C and 95% humidity, blotted for 1.5 s with blotting force −6. The grid was then flash frozen in liquid ethane and stored in the liquid nitrogen before data collection.

## Electron microscopy data collection

The cryoEM data were collected at eBIC (Diamond Light Source, UK) on a Titan Krios using EPU (FEI) operated at 300 kV and a K2 summit direct electron detector (Gatan). The data were collected with a defocus (underfocus) range between 1.5 µm to 3.5 µm. A total of 3543 micrographs were collected at pixel size of 1.06 Å/pixel and a dose of 50 $e^-/Å^2$ and each micrograph was fractioned into 41 frames (1.21 $e^-/Å^2$/frame).

## Image processing

The procedure of data processing is summarised in *Figure 2—figure supplement 1*. Frame alignment and dose weighting were carried out with MotionCor2 (*Zheng et al., 2017*) before estimating CTF parameters using Gctf (*Zhang, 2016*) and particle picking using Gautomatch (https://www.mrc-lmb.cam.ac.uk/kzhang/Gautomatch/) without a template. The picked particles were then extracted into boxes of 256 × 256 pixels. Initial 2D classification of the data was carried out in Cryosparc (*Punjani et al., 2017*) to remove junk particles due to ice contamination or other defects on the grids. Subsequent image processing was carried out in Relion 2.1 (*Scheres, 2012*). Briefly, the particles were separated using 3D classification. Three out of five classes were subsequently refined using the RNAP from the closed complex RPc (EMD-3695) filtered to 60 Å as the initial reference map. The remaining two classes were discarded due to their lack of clear structural features, probably derived from particles that are of poor quality. After combining the remaining three classes and refinement, one more round of focused 3D classification by applying a mask was carried out to separate different complexes or conformations. To generate the mask, structural models of RNAP and Ocr were first fitted into the refined 3D reconstruction. Ocr and the surrounding RNAP clamp regions were then used to calculate a volume in Chimera. Two classes with clearly different conformations of RNAP and both with Ocr bound, were refined, polished and post-processed (with masking and sharpening), resulting in the final reconstructions at 3.7 Å (for the wide clamp RNAP-Ocr) and 3.8 Å (for the narrow clamp RNAP-Ocr) based on the gold-standard Fourier shell correlation (FSC) at 0.143 criterion. The directional FSC analysis and histogram of directional resolution were performed using the 3D-FSC server (3dfsc.salk.edu) (*Tan et al., 2017*).

## Model building, refinement and structural analysis

The RNAP from RPo (PDB code: 6GH5) and RPip (PDB code: 6GH6) was used as an initial model for the model building of narrow clamp and wide clamp reconstructions, respectively. Briefly, the RNAP was first fitted into the RNAP-Ocr density map in Chimera (*Goddard et al., 2007*). Subsequently the RNAP structure was subject to flexible fitting using MDFF (*Trabuco et al., 2009*). The Ocr crystal structure (PDB code: 1S7Z) was manually fitted into the extra density of the RNAP-Ocr map in Coot (*Emsley et al., 2010*). Jelly body refinement in Refmac (*Murshudov et al., 2011*) and real space refinement in Phenix (*Afonine et al., 2012*) were used to improve the model quality. The final statistics of the model are in *Table 1*. The figures used for structure analysis and comparison were produced in Pymol (The PyMOL Molecular Graphics System, Version 2.0 Schrödinger, LLC) and UCSF Chimera (*Goddard et al., 2007*).

## Native gel mobility assay and western blotting

$σ^{54}$ as well as $σ^{70}$ were purified to homogeneity as described previously (*Nechaev and Severinov, 1999*; *Yang et al., 2015*). All the reactions were carried out in the binding buffer (20 mM Tris-HCl

**Table 1.** Cryo-EM data collection and refinement statistics.

| | RNAP-Ocr | |
|---|---|---|
| **Data collection and processing** | | |
| Magnification | 47393 | |
| Total micrographs | 3543 | |
| Movie frames | 41 | |
| Pixel size (Å) | 1.055 | |
| Defocus range (μm) | −1.2 to −3 | |
| Voltage (kV) | 300 | |
| Electron dose (e⁻/Å⁻²) $ (e^-/\text{Å}^{-2}) $ | 49.53 | |
| Total particles | 753783 | |
| FSC threshold | 0.143 | |
| **Reconstruction (RELION)** | **Wide clamp** | **Narrow clamp** |
| Particles | 33646 | 27312 |
| Resolution (Å) | 3.7 | 3.8 |
| **Refinement** | | |
| Resolution (Å) | 3.7 | 3.8 |
| **R.m.s. deviations** | | |
| Bond length (Å) | 0.003 | 0.003 |
| Bond angle (°) | 0.816 | 0.865 |
| **Ramachandran plot** | | |
| Favored regions (%) | 91.88 | 91.05 |
| Allowed regions (%) | 8.06 | 8.69 |
| Outlier | 0.06 | 0.27 |
| **Validation** | | |
| All-atom clashscore | 4.63 | 4.5 |
| Rotamer outliers (%) | 0.15 | 0.18 |
| C-beta deviations | 0 | 0 |

pH8.0, 200 mM KCl, 5 mM $MgCl_2$, 1 mM DTT, 5% glycerol) at room temperature, for the sequentially addition of different components in each step, the incubation time was approximately15-20 min and all the gels used were polyacrylamide. To test the effects of Ocr on holoenzyme formation for both $\sigma^{54}$ and $\sigma^{70}$, all the reactions were carried out in 20 μl at room temperature. The order of addition, final concentrations and molar ratio of different components are listed in corresponding figures. For each 20 μl reaction, 10 μl was taken out and loaded into one polyacrylamide gel (4.5%) and the remaining 10 μl of each sample was loaded to another identical gel (4.5%) and run in parallel for 160 mins at 80 volts at 4°C cold room. One gel was then stained with Coomassie blue and visualised, whilst the other gel was used for western blotting against the His-tagged protein of either $\sigma^{54}$ or $\sigma^{70}$, following the standard western blotting protocol (iBlot 2 Dry Blotting System, Thermo Fisher Scientific).

To test the effects of Ocr on the formation $\sigma^{70}$ open complexes, DNA probes used in this study were all Cy3 labelled. For $\sigma^{70}$ open complex experiments, one linear DNA used, the non-template DNA used was 5'- ACTTGACATCCCACCTCACGTATGCTATAATCCTACGAGTCTGACGCGG −3', and the template DNA used was 5'-CCGCGTCAGACTCGTAGGATTATAGCATACGTGAGGTGGGA TGTCAAGT −3'. All the other reactions were carried out at room temperature. The *lacUV5* linear DNA used was the same as reported (*Cámara et al., 2010*). The sequence of addition, the final concentration, molar ratios of different reactions are described in respective figures. All the gel (5%) are run at 4 °C in cold rooms, with 100 volts for 75 mins. The results were analysed by visualizing and quantifying the fluorescent signals of Cy3, attached to the DNA probes.

All the experiments were repeated at least once and the results are consistent with each other.

## Small primed RNA assays

All reactions were performed in 10 μl final volumes containing: STA buffer (*Burrows et al., 2010*), 100 nM holoenzyme (1:4 ratio of RNAP:σ$^{54}$), 20 nM promoter DNA probe (for σ$^{54}$ dependent transcription open complex formation, 4 mM dATP and 5 mM PspF$_{1-275}$ were also present) and incubated at room temperature. The sp RNA synthesis was initiated by adding 0.5 mM dinucleotide primer (ApG, ApA, CpA or UpG), 0.2 mCi/ml [α-$^{32}$P] GTP (3000 Ci/mmol) or 0.2 mCi/ml [α-$^{32}$P] UTP (3000 Ci/mmol). The reaction mixtures were quenched by addition of 4 μl of denaturing formamide loading buffer and run on a 20% denaturing gel and visualised using a Fuji FLA-5000 Phosphorimager. At least three independent experiments were carried out and values were within 5% of the relative % values measured.

## Bacterial growth curves

Growth of *E. coli* MG1655 cells not expressing Ocr (WT/pBAD18cm) and expressing Ocr (WT/pBAD18cm[*ocr*]) was monitored for 20 hr using a FLUOstar Omega plate reader (BMG LABTECH). Starting cultures of 0.1 OD$_{600}$ were inoculated in four different media, Luria-Bertani Broth (LB) with low (5 g/L) and high (10 g/L) salt concentration, nutrient broth (Oxoid) and modified M9 medium (Teknova), at two different temperatures, 37°C and 25°C. Expression of Ocr was induced using two different concentration of arabinose, 0.02%($^w$/$_v$) and 0.2%($^w$/$_v$).

## Beta-galactosidase assays

Gene expression levels of beta-galactosidase in *E. coli* MG1655 cells not expressing Ocr (WT/pBAD18cm) and expressing Ocr (WT/pBAD18cm[*ocr*]) were assessed using 0.002 mg/ml fluorescein di(beta-D-galactopyranoside) (FDG; Sigma-Aldrich). FDG results in fluorescence signals, which are proportional to the enzymatic activities after being hydrolysed to fluorescein by beta-galactosidase. Beta-galactosidase expression was induced by 1 mM IPTG 5 hr post-inoculation.

## RNA extraction, DNase digestion, reverse transcription and quantitative polymerase chain reaction (RT-qPCR)

Following the beta-galactosidase assays, total bacterial RNA was extracted from *E. coli* MG1655 cells using the RNeasy Mini Kit (Qiagen) according to the manufacturer's instructions. Residual DNA was digested by DNase I (Promega) and cDNA synthesis was performed using 500 ng of RNA and the SuperScript IV reverse transcriptase (Invitrogen) according to the manufacturer's instructions. Quantitation of the beta-galactosidase mRNA levels was performed using the specific oligonucleotide primers 5'-ATG GGT AAC AGT CTT GGC GG-3' and 5'-GGC GTA TCG CCA AAA TCA CC-3', the Power SYBR Green PCR Master Mix (Applied Biosystems) and the relative standard curve quantitation method as implemented by the OneStep Plus Real-Time qPCR System (Applied Biosystems).

## Acknowledgements

Initial screening was carried out at Imperial College London Centre for Structural Biology EM facility. High resolution data were collected at the eBIC, Diamond Light Source (proposal EM14769). eBIC is funded by the Wellcome Trust, MRC and BBSRC. This work is funded by the BBSRC to XZ and MB (BB/N007816). XL is funded by a China Scholarship Council studentship. We are especially grateful to David Ratner for information concerning his experiments.

## Additional information

### Funding

| Funder | Grant reference number | Author |
| --- | --- | --- |
| Biotechnology and Biological Sciences Research Council | BB/N007816/1 | Fuzhou Ye Milija Jovanovic Martin Buck Xiaodong Zhang |

| China Scholarship Council | Studentship | Xiaojiao Liu |
|---|---|---|

The funders had no role in study design, data collection and interpretation, or the decision to submit the work for publication.

## Author contributions
Fuzhou Ye, Data curation, Formal analysis, Validation, Visualization; Ioly Kotta-Loizou, Milija Jovanovic, Data curation, Formal analysis, Validation, Investigation, Visualization; Xiaojiao Liu, Data curation; David TF Dryden, Conceptualization, Resources, Methodology; Martin Buck, Resources, Supervision, Funding acquisition; Xiaodong Zhang, Conceptualization, Formal analysis, Supervision, Funding acquisition, Project administration

## Author ORCIDs
Martin Buck (iD) https://orcid.org/0000-0002-7580-8982
Xiaodong Zhang (iD) https://orcid.org/0000-0001-9786-7038

## Decision letter and Author response
Decision letter https://doi.org/10.7554/eLife.52125.sa1
Author response https://doi.org/10.7554/eLife.52125.sa2

# Additional files

## Supplementary files
• Supplementary file 1. Pairwise comparisons of *E. coli* cells expressing Ocr (WT/pBAD18cm[*ocr*]) *versus* cells not expressing Ocr (WT/pBAD18cm).

• Transparent reporting form

## Data availability
All data generated or analysed during the study are included in the manuscript and supporting files. The cryo EM maps and structural models will be deposited into EMDB and PDB with access codes 6R9G and 6R9B.

The following datasets were generated:

| Author(s) | Year | Dataset title | Dataset URL | Database and Identifier |
|---|---|---|---|---|
| Ye FZ, Zhang XD | 2020 | Structural basis of transcription inhibition by the DNA mimic Ocr protein of bacteriophage T7 | http://www.rcsb.org/structure/6R9G | RCSB Protein Data Bank, 6R9G |
| Ye FZ, Zhang XD | 2020 | Structural basis of transcription inhibition by the DNA mimic Ocr protein of bacteriophage T7 | http://www.rcsb.org/structure/6R9B | RCSB Protein Data Bank, 6R9B |

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
