## [Decision Letter]

**Acceptance summary:**

The results in this manuscript confirm that bacteriophage T7-encoded Ocr interacts with host *E. coli* RNAP, show that Ocr inhibits transcription by host RNAP, indicate that Ocr inhibits transcription by host RNAP through disruption of RNAP-σ and RNAP-DNA interactions, and define structures of RNAP-Ocr complexes that show Ocr interacts with the RNAP DNA binding determinants as a molecular mimic of DNA and suggest Ocr disrupts RNAP-σ and RNAP-DNA interactions through steric clash. The findings provide a striking example of protein function through structural mimicry of DNA. In addition, the results may provide an example of dual-targeted regulation – namely, a bacteriophage regulator that may functionally inhibit both host type I restriction endonucleases and host RNAP.

**Decision letter after peer review:**

Thank you for submitting your article "Structural basis of transcription inhibition by the DNA mimic protein Ocr of bacteriophage T7" for consideration by *eLife*. Your article has been reviewed by two peer reviewers, and the evaluation has been overseen by a guest Reviewing Editor and Cynthia Wolberger as the Senior Editor.

The reviewers have discussed the reviews with one another and the Reviewing Editor has drafted this decision to help you prepare a revised submission.

Summary:

Previous studies showed that bacteriophage T7 gp0.3 (Ocr) overcomes host restriction by serving as a molecular mimic of double-stranded DNA that binds to, and inhibits, host type I restriction endonucleases. Early work had also suggested that Ocr may interact with host RNAP. The results in this manuscript confirm that Ocr interacts with host RNAP and show that Ocr inhibits transcription by host RNAP by disrupting interactions with both σ factor and DNA interactions. The authors determine structures of RNAP-Ocr complexes, which show that Ocr interacts with the RNAP DNA binding determinants by serving as a molecular mimic of DNA and suggest Ocr disrupts RNAP-σ and RNAP-DNA interactions through steric clash. The findings provide a striking example of protein function through structural mimicry of DNA. In addition, the results may provide an example of dual-targeted regulation – namely, a bacteriophage regulator that may functionally inhibit both host type I restriction endonucleases and host RNAP.

Overall the structural work itself is of significant interest but requires substantial revision in terms of presentation. The part of the manuscript on mechanism is weaker but could be improved substantially with revisions (detailed below). The main weakness of the manuscript is that it does not sufficiently address to what extent the observed interaction of Ocr with RNAP is biologically relevant for bacteriophage T7 virulence, and how its functional importance compares to Ocr's interaction with type I restriction enzymes. The reviewers suggest that additional experiments addressing functional relevance is within the scope of the manuscript and are desirable but not essential.

Essential revisions:

I) Presentation of structural studies:

1) A broad issue concerning the presentation of the structural results is the overall focus on comparing Ocr-RNAP complexes with σ^54^-holoenzyme complexes since functional relevance of Ocr inhibition of σ^54^-dependent transcription is extremely unlikely. The reviewers agree that it's important to show the relationship between binding modes of Ocr in the Ocr-RNAP structures and binding modes of DNA in transcription initiation complexes, but that should be done with σ^70^-RNAP structures (which are all available). The presentation here makes it appear that phage-mediated inhibition of σ^54^-dependent transcription is a central component of phage infection, which is almost certainly not the case. The figures and data concerning Ocr inhibition of σ^54^-dependent transcription should be removed and/or moved to the supplement. These changes concern (although may not be limited to):

Figure 2C should show σ^70^-RPo complex, not σ^54^-RPo (6GH5).

Figure 4 should use a σ^70^-RPo, not σ^54^-RPo.

Data presented in Figure 6D, E – possibly irrelevant.

Data presented in Figure 7A, C – possibly irrelevant.

Data presented in Figure 8A – possibly irrelevant.

2) It is surprising that the authors did not compare the Ocr dimer structures in their cryo-EM maps with previously published Ocr structures, both in isolation and bound to EcoKI. Such a comparison, as well as of the interactions of Ocr with RNAP and EcoKI, is warranted at least as a comment in the main text, if not in a figure.

3) There are many issues regarding the figures and presentation of structural and mechanistic data. These are detailed below (Figure revisions).

II) Studies of Ocr inhibition mechanism:

1) The data on transcription inhibition in uninfected cells engineered to overproduce Ocr (Figure 5) is relatively uninformative and should be removed.

2) Figure 6, presenting confusing 'reaction schematics' and gel images for 44 different assays and conditions should be replaced by bar graphs with error bars, presenting the most important assays and conditions.

3) The in vitro transcription experiments of Figure 6 were performed with fully duplex promoter DNAs, while the gel shifts and competition experiments of Figure 7C and 7D are done with promoter DNAs that have mismatch transcription bubbles – that explains why Ocr reduces in vitro transcription from 100 to 39%, but in 7D (for example), there is no decrease in open complex formation whatsoever. It’s not clear why the experiments were done this way – the gel shift/competition experiments with mismatch promoters aren't really very informative of anything. These experiments should be removed or repeated with relevant DNA templates (i.e. fully duplex).

4) The text describing the Ocr inhibition mechanism could be shortened.

III) Biological relevance of inhibition of RNAP function by Ocr to T7 phage virulence:

The weakest part of the manuscript is the proposal that Ocr functionally inhibits host RNAP during infection by bacteriophage T7 – i.e., the proposal that inhibition of host RNAP by OCR occurs during, and is important for, infection by bacteriophage T7. This proposal should be either: (1) tempered with strong caveats, or (2) bolstered by additional experimental data. One experiment that potentially could be performed quickly would be to compare burst sizes for infection of host cells lacking type I restriction endonucleases (available commercially) with bacteriophage T7 derivatives having or lacking the gene encoding Ocr; If the authors proposal is true, there should be measurable differences in burst sizes. The reviewers strongly encourage, but do not absolutely require, the authors to attempt to compare burst sizes of ocr+ and ocr= T7 phage in hosts strains lacking type I restriction.

Figure Revisions:

Figure 1:

1) In A, there are two peaks of the RNAP-Ocr complex eluting between 40-57 ml. Do the authors know what the difference is between the complexes in both peaks? Which peak was used for structural studies?

2) In C, it is unclear why the RNAP-Ocr complex is loaded twice in the western blot. What's the difference between both lanes?

3) In D-E, the authors use EC50 and Kd interchangeably. To avoid confusion, the authors should be consistent and refer to Kd. Moreover, considering the labeled ligand in these experiments is at 25 nM concentration and the Kd for the interactions is in a similar concentration range, the binding measurements are performed under conditions of RNAP depletion (i.e. the free concentration of RNAP does not equal the concentration of added RNAP for each data point). If the authors cannot use ligand concentration 10-fold below the Kd for technical reasons, they should use a binding model that accounts for this change in free RNAP concentration or at least acknowledge that the reported Kds are higher than the true Kds.

4) The authors also state that Ocr is much more abundant in T7 infected cells than type I restriction enzymes, arguing that the additional Ocr molecules interact with PNAP. It would be helpful if the authors could comment on the intracellular concentration of Ocr compared to σ factors and whether these are in a range where Ocr can effectively compete with σ factors for binding to RNAP.

Figure 2:

5) Should be revised to compare Ocr-RNAP binding with σ^70^-holoenzyme transcription initiation complexes.

6) For non-experts, it is not easily evident where the template and non-template DNA strands, as well as the RNA, are located in the structure. The authors should consider coloring these elements differently (also in Figure 3).

Figure 2—figure supplement 1:

7) The authors used focused 3D classification but do not explain how the masks were generated. A description should be included in the Materials and methods section, and it would be useful to also show the masks used in the figure.

Figure 2—figure supplement 2:

8) A and B, the particle orientation distributions are far from isotropic, the effect of this on the FSC curves should be assessed with directional FSC plots (https://3dfsc.salk.edu/). Some of the density maps shown in Figure 2—figure supplement 3 may suffer from this.

9) In A and B, the display threshold for rendering the EM volumes seems too low. Increasing the threshold will likely help to bring out the secondary structure features that should be easily identifiable at this resolution but are not well visible in the representations shown in the figures.

10) In D, what's the difference between both half-map FSCs? Presumably one map was used for model refinement and the other as control, but it is not clear which is which.

Figure 2—figure supplement 3:

11) The authors state that many side chain densities are resolved in the map, but this is not well visible in panel C.

Figure 4:

12) Should be revised to compare Ocr-RNAP binding with σ^70^-holoenzyme transcription initiation complexes.

13) The authors state in the text (subsection “Structures of Ocr in complex with RNAP”, last paragraph) that the downstream DNA binding channel provides a positively charged surface composed of b, b' clamps and b' jaw, but this is not shown in the figure. The authors further state that positively charged areas on the Ocr surface face the b' jaw domain, but this is also not evident in this figure as the b' jaw is not shown in B. It would be helpful to revise the figure accordingly.

Figure 7 and Figure 7—figure supplement 1:

14) How were the gels stained in the different panels?

15) In Figure 7A and B, the authors argue that there is a gradual disappearance of the RNAP-σ^54^ complex when adding increasing concentrations of Ocr. However, in conditions I and II in 7A there seems to be no or very little RNAP-σ^54^ complex present at either Ocr concentration. A more careful titration would be necessary to make this point. The low abundance of RNAP-σ^54^ in lanes 7 and 9 is also surprising since one would expect a similar abundance of both RNAP-Ocr and RNAP-σ^54^ at equimolar concentration of the components given the similar Kds for the interactions. In condition III, the authors add Ocr to preformed RNAP-σ^54^ purified by gel filtration, but it seems there is barely any σ^54^ present (see also the very weak signal in Figure 7—figure supplement 1A when comparing lanes 11 and 12 to lane 6). Therefore, it is unclear whether Ocr actually competes with σ^54^ or binds to abundant free RNAP. To substantiate their claim, the authors should demonstrate that the preformed RNAP-σ complexes contain stoichiometric amounts of all components as expected. More generally, the gels shown could also benefit from quantitation.

16) In Figure 7C and D, the authors changed the protein concentrations for the different reactions compared to A and B. To ease comparison with the other panels in this figure, it would make more sense to use similar experimental conditions as in the previous experiments. It would also be helpful if the authors could comment on the affinity of the DNA for RNAP-σ. Is binding tighter than that of Ocr to RNAP to explain that DNA can overcome the inhibitory effect of Ocr that is pre-bound to RNAP?

[Editors' note: further revisions were suggested prior to acceptance, as described below.]

Thank you for resubmitting your work entitled "Structural basis of transcription inhibition by the DNA mimic protein Ocr of bacteriophage T7" for further consideration by *eLife*. Your revised article has been evaluated by Cynthia Wolberger (Senior Editor) and a Reviewing Editor.

The manuscript has been improved and the comments of the reviewers have been satisfactorily addressed. However, there is a remaining issues with one of the figures that needs to be addressed before acceptance, as outlined below:

Figure 6 in the revised manuscript is problematic – the discussion of Figure 6 in the text, in the Figure 6 legend, and the figure itself are inconsistent. While Figure 6 contains panels A – D, the figure legend only refers to panels A – C. In addition, the legend does not adequately describe the figures. The legend to panel A should describe the experiment more completely and explain what Roman numerals I, II and III denote in that panel, as well as in panel B. It would perhaps be best to relabel the panels to denote the reaction scheme as panel A and to explicitly point the reader to that scheme in explaining I, II and III. The differences between the pairs of lanes 7-8, 9-10, 11-12 should also be explained.

[Editors' note: further revisions were suggested prior to acceptance, as described below.]

Thank you for submitting your article "Structural basis of transcription inhibition by the DNA mimic protein Ocr of bacteriophage T7" for consideration by *eLife*. The evaluation has been overseen by a Reviewing Editor and Cynthia Wolberger as the Senior Editor.

Figure 6 and the legend are now improved in that one can better understand what is in the figure based on the legend. In panels C and D, the authors helpfully added labels above the gel lanes to explain the differences between the pairs of lanes in 'I', 'II', and 'III' (0.5 vs. 5 microM); however, the same needs to be done for panels A and B. Please label panels A and B with the concentrations of Ocr above the pairs of lanes in I, II, and III so that all the panels in the figure are labeled optimally and the same.

---

## [Author Response]

Essential revisions:I) Presentation of structural studies:1) A broad issue concerning the presentation of the structural results is the overall focus on comparing Ocr-RNAP complexes with σ^54^-holoenzyme complexes since functional relevance of Ocr inhibition of σ^54^-dependent transcription is extremely unlikely. The reviewers agree that it's important to show the relationship between binding modes of Ocr in the Ocr-RNAP structures and binding modes of DNA in transcription initiation complexes, but that should be done with σ^70^-RNAP structures (which are all available). The presentation here makes it appear that phage-mediated inhibition of σ^54^-dependent transcription is a central component of phage infection, which is almost certainly not the case. The figures and data concerning Ocr inhibition of σ^54^-dependent transcription should be removed and/or moved to the supplement.

σ^54^-dependent transcription plays vital roles in responding to phage infection. In fact, our model system for σ^54^-dependent transcription is the psp (phage shock protein) system, which is induced by filamentous phage infection among others. However, we have no data in supporting whether σ^54^-dependent transcription is a central component here and have replaced the figures with σ^70^ and moved σ^54^ comparisons, if relevant, to supplementary data.

These changes concern (although may not be limited to):Figure 2C should show σ^70^-RPo complex, not σ^54^-RPo (6GH5).

We have done so.

Figure 4 should use a σ^70^-RPo, not σ^54^-RPo.

We have done so.

Data presented in Figure 6D, E – possibly irrelevant.

It is clear that Ocr inhibits both σ^54^ and σ^70^-dependent transcription in vitro although it is unclear the exact in vivo effect. We think the data should be presented but have moved σ^54^-dependent data to supplementary.

Data presented in Figure 7A, C – possibly irrelevant.

Again we have decided to present the data but moved to supplementary.

Data presented in Figure 8A – possibly irrelevant.

We have removed the direct comparison with RNAP-σ^54^ structure.

2) It is surprising that the authors did not compare the Ocr dimer structures in their cryo-EM maps with previously published Ocr structures, both in isolation and bound to EcoKI. Such a comparison, as well as of the interactions of Ocr with RNAP and EcoKI, is warranted at least as a comment in the main text, if not in a figure.

The resolution for Ocr is insufficient for building an atomic model so we used the Ocr crystal structure as the starting model and then fitted into the density and carried out refinement. Overall there are no significant differences at the monomeric level between the Ocr model in this study and that of crystal structure and the dimer interface is also largely unchanged. However the dimer is slightly less bent compared to that of crystal structure. We have added a sentence to describe this and a supplementary figure (Figure 2—figure supplement 4). In terms of EcoKI complex, the structural model was derived from a negatively stained reconstruction and the reliability and accuracy are questionable hence we are reluctant to perform a detailed comparison. Nevertheless, we have added a sentence stating that in that complex, Ocr is fully encircled by the enzyme, similar to DNA encircled by the DNA clamp while Ocr in the transcriptional complex is bound in the cleft.

3) There are many issues regarding the figures and presentation of structural and mechanistic data. These are detailed below (Figure revisions).II) Studies of Ocr inhibition mechanism:1) The data on transcription inhibition in uninfected cells engineered to overproduce Ocr (Figure 5) is relatively uninformative and should be removed.

Here we show that expression of Ocr in vivo inhibited transcription in a test reporter assay, consistent with in vitro data and suggesting that Ocr could perform as an antimicrobial. We think this is relevant information and have kept the main data in Figure 5 (Figure 5C-D) but reduced the text to describe these. We have also moved some of the data to new Figure 5—figure supplement 2.

2) Figure 6, presenting confusing 'reaction schematics' and gel images for 44 different assays and conditions should be replaced by bar graphs with error bars, presenting the most important assays and conditions.

We have added bar graphs as suggested and moved σ^54^-related data to supplementary (new Figure 5—figure supplement 1).

*3) The* in vitro *transcription experiments of Figure 6 were performed with fully duplex promoter DNAs, while the gel shifts and competition experiments of Figure 7C and D are done with promoter DNAs that have mismatch transcription bubbles – that explains why Ocr reduces* in vitro *transcription from 100 to 39%, but in 7D (for example), there is no decrease in open complex formation whatsoever. It's not clear why the experiments were done this way – the gel shift/competition experiments with mismatch promoters aren't really very informative of anything. These experiments should be removed or repeated with relevant DNA templates (i.e. fully duplex).*

We have now repeated these experiments with fully duplex DNA and results are shown in new Figure 6 and have quantified and added bar graphs. We performed mismatched DNA for both σ^54^ and σ^70^ to compare effects on open complex, which turned out to be negligible. We have now removed these data. The competition experiments for promoter complex formation (Figure 6) are now under the same conditions as those of in vitro transcription assays (Figure 5) and indeed they agree very well with each other.

4) The text describing the Ocr inhibition mechanism could be shortened.

We have now shortened this part.

III) Biological relevance of inhibition of RNAP function by Ocr to T7 phage virulence:The weakest part of the manuscript is the proposal that Ocr functionally inhibits host RNAP during infection by bacteriophage T7 – i.e., the proposal that inhibition of host RNAP by OCR occurs during, and is important for, infection by bacteriophage T7. This proposal should be either: (1) tempered with strong caveats, or (2) bolstered by additional experimental data. One experiment that potentially could be performed quickly would be to compare burst sizes for infection of host cells lacking type I restriction endonucleases (available commercially) with bacteriophage T7 derivatives having or lacking the gene encoding Ocr; If the authors proposal is true, there should be measurable differences in burst sizes. The reviewers strongly encourage, but do not absolutely require, the authors to attempt to compare burst sizes of ocr+ and ocr= T7 phage in hosts strains lacking type I restriction.

We appreciate the suggested experiments and in fact earlier experiments carried out by McAllister and Barrett, 1977, to some extent have addressed this issue. In that study, the authors used host strains lacking the host restriction enzymes and looked at the effects of host RNA and protein synthesis post T7 infection. Their results showed that expression of gp0.3 (Ocr) resulted in an inhibition, albeit delayed, of host β-galactosidase synthesis and delayed inhibition of total RNA synthesis (Figure 3 in McAllister and Barrett, 1977). This inhibitory effect is consistent with our results here showing that over-expressed Ocr inhibits host transcription, and thus also protein production. The delayed response could be due to the fact that Ocr is expressed by the host RNAP, it thus takes time for sufficient amount of Ocr to be synthesized by the host to show an inhibitory response. We have added this in the text. Interestingly, when Ocr and gp1.1 (a small protein with unknown function) were both expressed in T7, the host protein synthesis was shutdown effectively, suggesting that gp1.1 and Ocr might have complementary/synergetic effects although that requires further investigation. We have added this in Discussion but have also tuned down on the proposal.

Figure Revisions:Figure 1:1) In A, there are two peaks of the RNAP-Ocr complex eluting between 40-57 ml. Do the authors know what the difference is between the complexes in both peaks? Which peak was used for structural studies?

Both peaks contain RNAP-Ocr as indicated by SDS-PAGE. They could reflect the two distinct conformations we observe. The peaks were pulled together.

2) In C, it is unclear why the RNAP-Ocr complex is loaded twice in the western blot. What's the difference between both lanes?

They are fractions from the two peaks, again showing both containing Ocr. We have marked this and made clear in figure legend.

3) In D-E, the authors use EC50 and Kd interchangeably. To avoid confusion, the authors should be consistent and refer to Kd. Moreover, considering the labeled ligand in these experiments is at 25 nM concentration and the Kd for the interactions is in a similar concentration range, the binding measurements are performed under conditions of RNAP depletion (i.e. the free concentration of RNAP does not equal the concentration of added RNAP for each data point). If the authors cannot use ligand concentration 10-fold below the Kd for technical reasons, they should use a binding model that accounts for this change in free RNAP concentration or at least acknowledge that the reported Kds are higher than the true Kds.

We appreciate and agree with the reviewers on these points. Indeed, the signal from the labelled protein is a limiting factor on the concentrations we could use in these experiments. We have now refitted the data using the full tight binding equation. A tight binding equation was used with σ or Ocr fixed at 25 nM. The Kd values now are Ocr-RNAP: 7 nM, σ^70^-RNAP: 15 nM and σ^54^-RNAP: 20 nM. These numbers are broadly in agreement with Kd of σ^54^-RNAP (~ 60 nM) obtained in our laboratory (Scott et al. Biochem J., 2000, 352:539-547) using kinetic binding experiments. The competition experiments in Figure 6 (and Figure 6—figure supplement 1) show that equimolar concentration of Ocr can effectively compete out σ^70^ and σ^54^, again consistent with similar Kds to RNAP. Nevertheless, we have made the description more qualitative in the text and made it clear that the Kd values represent maximum values.

4) The authors also state that Ocr is much more abundant in T7 infected cells than type I restriction enzymes, arguing that the additional Ocr molecules interact with PNAP. It would be helpful if the authors could comment on the intracellular concentration of Ocr compared to σ factors and whether these are in a range where Ocr can effectively compete with σ factors for binding to RNAP.

It is difficult to pin down exact numbers but RNAP is estimated to be ~1000 and is in excess of σ^70^ (~ 500) and σ^54^ (< 100) (see https://pax-db.org/dataset/511145/137/). Based on experiments conducted in one of the authors (DFTD)’s lab in terms of SDS-PAGE gel intensities, there are at least a few hundreds of Ocr molecules, compared to < 60 restriction enzymes in infected cells.

Figure 2:5) Should be revised to compare Ocr-RNAP binding with σ^70^-holoenzyme transcription initiation complexes.

We have done so.

6) For non-experts, it is not easily evident where the template and non-template DNA strands, as well as the RNA, are located in the structure. The authors should consider coloring these elements differently (also in Figure 3).

We have now colored them differently.

Figure 2—figure supplement 1:7) The authors used focused 3D classification but do not explain how the masks were generated. A description should be included in the Materials and methods section, and it would be useful to also show the masks used in the figure.

We have now added the masks in the figures and detailed how the mask is generated.

Figure 2—figure supplement 2:8) A and B, the particle orientation distributions are far from isotropic, the effect of this on the FSC curves should be assessed with directional FSC plots (https://3dfsc.salk.edu/). Some of the density maps shown in Figure 2—figure supplement 3 may suffer from this.

Yes the dataset does suffer from preferential orientation issues and the directional FSC plots confirm that. We thank the reviewers for recommendations. We have conducted the analysis and included these plots in Figure 2—figure supplement 2. Furthermore, we have discussed these issues in the text.

9) In A and B, the display threshold for rendering the EM volumes seems too low. Increasing the threshold will likely help to bring out the secondary structure features that should be easily identifiable at this resolution but are not well visible in the representations shown in the figures.

We have replaced the figures.

10) In D, what's the difference between both half-map FSCs? Presumably one map was used for model refinement and the other as control, but it is not clear which is which.

Yes the reviewers are correct and we have now clearly labelled and explained in figure legends.

Figure 2—figure supplement 3:11) The authors state that many side chain densities are resolved in the map, but this is not well visible in panel C.

We have replaced with new figures to better reflect the quality in the core region.

Figure 4:12) Should be revised to compare Ocr-RNAP binding with σ^70^-holoenzyme transcription initiation complexes.

We have done so.

13) The authors state in the text (subsection “Structures of Ocr in complex with RNAP”, last paragraph) that the downstream DNA binding channel provides a positively charged surface composed of b, b' clamps and b' jaw, but this is not shown in the figure. The authors further state that positively charged areas on the Ocr surface face the b' jaw domain, but this is also not evident in this figure as the b' jaw is not shown in B. It would be helpful to revise the figure accordingly.

We have made new figures (Figure 4) to illustrate these features.

Figure 7 and Figure 7—figure supplement 1:14) How were the gels stained in the different panels?

Some were stained for proteins using Coomassie blue, some were western blots (anti-his tag) and visualized by chemical colorimetric exposing. Some were visualized by detecting the Cy3 flourophore signals covalently linked to DNA. This information is now provided in Figure Legends and/or Materials and methods.

15) In Figure 7A and B, the authors argue that there is a gradual disappearance of the RNAP-σ^54^ complex when adding increasing concentrations of Ocr. However, in conditions I and II in 7A there seems to be no or very little RNAP-σ^54^ complex present at either Ocr concentration. A more careful titration would be necessary to make this point. The low abundance of RNAP-σ^54^ in lanes 7 and 9 is also surprising since one would expect a similar abundance of both RNAP-Ocr and RNAP-σ^54^ at equimolar concentration of the components given the similar Kds for the interactions. In condition III, the authors add Ocr to preformed RNAP-σ^54^ purified by gel filtration, but it seems there is barely any σ^54^ present (see also the very weak signal in Figure 7—figure supplement 1A when comparing lanes 11 and 12 to lane 6).

The Kd for Ocr is actually slightly better than for both σ^70^ and σ^54^ (Figure 1D-F). Furthermore, σ does not stain so well. They are more visible in western blotted gels against His-tag on σ proteins (new Figure 6B). We have also moved σ^54^ data to supplementary figures (new Figure 6—figure supplement 1).

Therefore, it is unclear whether Ocr actually competes with σ^54^ or binds to abundant free RNAP. To substantiate their claim, the authors should demonstrate that the preformed RNAP-σ complexes contain stoichiometric amounts of all components as expected. More generally, the gels shown could also benefit from quantitation.

We have shown that the holoenzyme is formed stoichiometrically (Figure 1—figure supplement 1). We have quantified gels when possible.

16) In 7C and 7D, the authors changed the protein concentrations for the different reactions compared to A and B. To ease comparison with the other panels in this figure, it would make more sense to use similar experimental conditions as in the previous experiments. It would also be helpful if the authors could comment on the affinity of the DNA for RNAP-σ. Is binding tighter than that of Ocr to RNAP to explain that DNA can overcome the inhibitory effect of Ocr that is pre-bound to RNAP?

We have conducted experiments under the same conditions as in vitro transcription assays and stained for Cy3 fluorescently labelled DNA (new Figure 6C-D compared to new Figure 5). Figure 6A-B shows competition experiments and protein concentrations are 2x as in Figure 6C-D due to different detection methods. In Figure 6A, the gels were stained with Coomassie blue. However σ factors were not sensitive to Coomassie. Figure 6B shows western blots against His-tag on σ. Figure 6C-D investigates the effects on promoter complex formation, similar to those in Figure 5. Here we used fluorescent signals from Cy3 attached to DNA, which are more sensitive than Coomassie and hence lower protein concentrations were used. We now also include data showing DNA binding to RNAP-σ^70^ and indeed the Kd is ~ 0.1 nM, significantly tighter binding compared to Ocr. These data are included in Figure 6—figure supplement 1.

[Editors' note: further revisions were suggested prior to acceptance, as described below.]

Figure 6 in the revised manuscript is problematic – the discussion of Figure 6 in the text, in the Figure 6 legend, and the figure itself are inconsistent. While Figure 6 contains panels A – D, the figure legend only refers to panels A – C. In addition, the legend does not adequately describe the figures. The legend to panel A should describe the experiment more completely and explain what Roman numerals I, II and III denote in that panel, as well as in panel B. It would perhaps be best to relabel the panels to denote the reaction scheme as panel A and to explicitly point the reader to that scheme in explaining I, II and III. The differences between the pairs of lanes 7-8, 9-10, 11-12 should also be explained.

We have now amended Figure 6 legend as well as the text (subsection “Ocr inhibits RNAP recruitment and open complex formation but does not 271 disrupt pre-formed open complexes”) in the revised manuscript. I hope now the manuscript can be accepted for publication in *eLife*.

[Editors' note: further revisions were suggested prior to acceptance, as described below.]

Figure 6 and the legend are now improved in that one can better understand what is in the figure based on the legend. In panels C and D the authors helpfully added labels above the gel lanes to explain the differences between the pairs of lanes in 'I', 'II', and 'III' (0.5 vs. 5 microM); however, the same needs to be done for panels A and B. Please label panels A and B with the concentrations of Ocr above the pairs of lanes in I, II, and III so that all the panels in the figure are labeled optimally and the same.

We have now amended Figure 6 as well as Figure 6—figure supplement 1 with Ocr concentrations clearly labelled above the pairs of lanes.